# Interaction Between PHF8 and a Segment of KDM2A, Which Is Controlled by the Phosphorylation Status at a Specific Serine in an Intrinsically Disordered Region of KDM2A, Regulates rRNA Transcription and Cell Proliferation in a Breast Cancer Cell Line

**DOI:** 10.3390/biom15050661

**Published:** 2025-05-02

**Authors:** Kengo Okamoto, Yutaro Mihara, Sachiko Ogasawara, Takashi Murakami, Sinya Ohmori, Tetsuya Mori, Toshiyuki Umata, Yuki Kawasaki, Kazuya Hirano, Hirohisa Yano, Makoto Tsuneoka

**Affiliations:** 1Faculty of Agriculture, Department of Applied Biological Science, Takasaki University of Health and Welfare, Takasaki 370-0033, Gunma, Japan; kokamoto@takasaki-u.ac.jp; 2Department of Pathology, School of Medicine, Kurume University, Kurume 830-0011, Fukuoka, Japan; mihara_yuutarou@med.kurume-u.ac.jp (Y.M.); sachiko@med.kurume-u.ac.jp (S.O.); hiroyano@med.kurume-u.ac.jp (H.Y.); 3Department of Microbiology, Faculty of Medicine, Saitama Medical University, Iruma 350-1241, Saitama, Japan; takmu@saitama-med.ac.jp; 4Faculty of Pharmacy, Takasaki University of Health and Welfare, Takasaki 370-0033, Gunma, Japan; omori@takasaki-u.ac.jp (S.O.); tmori@takasaki-u.ac.jp (T.M.); kawasaki-y@takasaki-u.ac.jp (Y.K.); hirano-k@takasaki-u.ac.jp (K.H.); 5Radioisotope Research Center, Facility for Education and Research Support, University of Occupational and Environmental Health, Kitakyushu 807-8555, Fukuoka, Japan; umata@med.uoeh-u.ac.jp

**Keywords:** KDM2A, PHF8, IDR, AMPK, rRNA transcription, 2DG, dephosphorylation

## Abstract

Mild starvation due to low concentrations of an inhibitor of glycolysis, 2-deoxy-D-glucose, activates AMP-activated protein kinase (AMPK) and lysine-specific demethylase 2A (KDM2A) to reduce rRNA transcription and cell proliferation in breast cancer cells. However, the mechanisms of how AMPK regulates KDM2A are unknown. Here, we found that PHD finger protein 8 (PHF8) interacted with KDM2A and contributed to the reduction in rRNA transcription and cell proliferation by 2-deoxy-D-glucose in a breast cancer cell line, MCF-7. We analyzed how KDM2A bound PHF8 in detail and found that PHF8 interacted with KDM2A via two regions of KDM2A. One of the regions contained an intrinsically disordered region (IDR). IDRs can show rapidly switchable protein–protein interactions. Deletion of the PHF8-binding region activated KDM2A to reduce rRNA transcription, and 2-deoxy-D-glucose reduced the interaction between PHF8 and the KDM2A fragment containing the PHF8-binding region. A 2-deoxy-D-glucose or AMPK activator dephosphorylated KDM2A at Ser731, which is located on the N-terminal side of the PHF8-binding region. Replacement of Ser731 by Ala decreased binding of PHF8 to the KDM2A fragment that contains the PHF8-binding region and Ser731 and reduced rRNA transcription and cell proliferation. These results suggest that the mode of interaction between KDM2A and PHF8 is regulated via dephosphorylation of KDM2A through AMPK to control rRNA transcription, and control of the phosphorylation state of Ser731 would be a novel target for breast cancer therapy.

## 1. Introduction

The level of rRNA transcription is controlled by environmental conditions and largely affects cell proliferation [1,2,3,4]. To date, increasing numbers of studies have revealed that signals from the environment reach the rRNA transcription machinery in a nucleolus to regulate rRNA transcription [1,2,3,4,5,6,7,8,9]. The lysine-specific histone demethylase KDM2A belongs to the KDM2 family [10], which was the first identified JmjC-type histone demethylase. KDM2A is a modular protein containing JmjC, CxxC-ZF, plant homeodomain (PHD), F-box, and leucine-rich repeats (LRRs) domain structures involved in regulation of transcription [10]. Because the rRNA gene promoter contains lots of CpG dinucleotides [11,12] and the CxxC-ZF domain binds to unmethylated CpG [11,13,14,15], we previously tested whether the CxxC-ZF domain was involved in the binding of KDM2A to the rRNA gene promoter [9]. The recombinant GST-fusion protein containing the CxxC-ZF domain of KDM2A binds to the rRNA gene promoter depending on the unmethylated state of DNA and the integrity of the CxxC-ZF domain in vitro. The binding of KDM2A to the rRNA gene promoter in cells was also observed by ChIP analysis in vivo. These results indicate that KDM2A is recruited to the rRNA gene promoter depending on the CxxC-ZF domain. KDM2A is expressed throughout the body during embryogenesis, and studies with *Kdm2a*-knockout mice suggest that KDM2A plays an essential role in embryonic development in higher animals [16]. While some reports suggest that KDM2A affects cancer cell proliferation positively [17,18,19,20,21,22], KDM2A was also shown to affect cell proliferation negatively [23,24]. Under starvation conditions, rRNA transcription is reduced, which is highly dependent on KDM2A in low concentrations of 2-deoxy-D-glucose (2DG), an inhibitor of glycolysis. This mild starvation condition also reduces proliferation of breast cancer cells in a KDM2A-dependent manner [23,24]. These results suggest that KDM2A fine-tunes rRNA transcription. 2DG activates AMP-activated protein kinase (AMPK) and activates KDM2A to reduce rRNA transcription [23,24]. AMPK is a key metabolic sensor that has a pivotal role in the maintenance of cellular energy homeostasis [25,26]. However, the mechanisms of how AMPK controls KDM2A to reduce rRNA transcription remain to be clarified.

Intrinsically disordered regions (IDRs) are segments in a protein sequence that lack stable structures under physiological conditions [27,28,29,30,31,32]. IDRs play important roles in many cellular activities, complementing the functions of structured proteins and domains [27,31]. While proteins composed mostly of folded domains are enriched among metabolic enzymes and proteins regulating homeostasis, proteins abundant in IDRs are common among factors that regulate chromatin structures or transcription [28]. The roles of IDRs in transcription regulation appear to be generally related to their characters, including organizing rapidly switchable protein–protein interactions [28]. Recently, the molecular details that underlie how individual IDRs with distinct amino acid sequences confer different biological functions have begun to be described in some biological contexts [29,30]. However, the information linking IDRs to specific molecular functions was not reported in many cases [32].

To investigate the mechanisms that regulate the activity of KDM2A in the nucleolus, we previously searched for amino acid sequences involved in nucleolar localization [24]. The results showed that there are multiple regions involved in nucleolar localization of KDM2A. One region contains CxxC-ZF, which directly binds to the rRNA gene promoter [9], and another region binds directly to heterochromatin protein 1γ (HP1γ) using valine 801 in the LxVxL motif of KDM2A [24]. Both regions were required for KDM2A to suppress rRNA transcription during starvation [9,24]. To investigate the mechanisms controlling KDM2A activities in the nucleolus, further KDM2A-interacting proteins were sought, and PHF8 was found to bind KDM2A. PHF8 acts as a demethylase for the repressive histone marks [33]. PHF8 has six putative nuclear localization signals in its amino acid sequence [34]. PHF8 binds the active histone marks through its PHD. Recombinant PHF8 bound to a synthetic histone H3 peptide comprising H3K4me3 and, to a lesser extent, to H3K4me2, but did not interact with unmodified histone H3 peptide, H3K4me1, H3K9me2, or H3K9me3 [35]. These results suggest that PHF8 is anchored to the active rDNA repeats that are demarcated by H3K4me3. PHF8 up-regulates rRNA transcription by reducing the repressive histone marks (H3K9me2) through its demethylase activity [35,36]. However, whether PHF8 activity is altered during starvation has not yet been tested. Prior studies have shown that mutations in PHF8 are associated with X-linked mental retardation and cleft lip/cleft palate [34,37,38,39]. Interestingly, recent studies have revealed that PHF8 is aberrant in several human malignancies [40], including breast cancer [41,42,43].

In this paper, we reported a molecular mechanism for regulation of KDM2A during starvation. We found that KDM2A bound PHF8 through two distinct regions of KDM2A. While five IDRs exist in the KDM2A protein [44], one of the PHF8-interacting regions contains an IDR. It was found that the IDR-containing region showed a switchable interaction with PHF8 and played a role in controlling rRNA transcription.

## 2. Materials and Methods

### 2.1. Cells and Cell Culture

Cells were cultured at 37 °C in an atmosphere containing 5% CO_2_ and 100% humidity. The human breast adenocarcinoma cell line MCF-7 (Cell Resource Center for Biomedical Research, Tohoku University, Sendai, Miyagi, Japan, TKG 0479::MCF7) was cultured in RPMI1640 medium (Nacalai Tesque, Kyoto, Japan) supplemented with 10% fetal calf serum (FCS). The human embryonic kidney (HEK) 293 cell line expressing the large T antigen of simian virus 40 (SV40), 293T cells (RIKEN Cell Bank, Tsukuba, Ibaraki, Japan, RCB2202:293T), was cultured in Dulbecco’s modified Eagle’s medium (DMEM, Cat# D5796, Sigma-Aldrich Co., St. Louis, MO, USA) supplemented with 10% FCS.

### 2.2. Introduction of Plasmids and siRNAs into Mammalian Cells

Confluent cells were replated with fresh medium and cultured for two or three days. Mammalian expression plasmids were introduced into cells using FuGENE6 transfection reagent (Promega, Madison, WI, USA) or super electroporation NEPA21 type II (Nepa Gene Co., Ltd., Chiba, Japan), according to the manufacturers’ instructions. Cells were transfected with stealth small interfering RNA (siRNA) using Lipofectamine RNAiMAX (Thermo Fisher Scientific, Waltham, MA, USA) according to the manufacturer’s instructions after cells were replated. The siRNA oligonucleotide sequences used are shown below (Table 1).

### 2.3. Plasmids

KDM2A was expressed under a pCAGGS mammalian expression vector or a pCAGGS-Flag mammalian expression vector, which were described previously [8,9,23]. The cDNAs encoding mutant KDM2A were constructed using a KOD-Plus Mutagenesis Kit, a polymerase chain reaction (PCR)-based mutation-introducing system (Toyobo, Osaka, Japan), according to the manufacturer’s instructions. The cDNA encoding a mutant KDM2A whose expression was resistant to the siKDM2A was described previously [8,9,24]. After confirming their sequences, cDNAs were subcloned into the mammalian expression vectors. The plasmids expressing green fluorescent protein (GFP)-fusion KDM2A mutants were constructed in a pEGFP expression vector (TaKaRa Bio Inc., Ohtsu, Japan) by PCR amplification with appropriate primers. pFN21AA1111, which encodes Halo tag-PHF8 protein, was obtained from Kazusa Research Institute. The cDNA encoding PHF8 was subcloned into a pCAGGS mammalian expression vector or a pCAGGS-Flag mammalian expression vector [8].

### 2.4. Western Blotting of Cell Lysates and Immunoprecipitation

For Western blotting, cells were trypsinized and extracted in 3% SDS solution containing 100 mM Tris-HCl, pH 6.8, 0.05 M DTT, and 20% glycerol (3% SDS solution). Cell extracts were separated by SDS-polyacrylamide gel electrophoresis (SDS-PAGE) and transferred to a microporous PVDF membrane (Merck Millipore, Darmstadt, Germany). After treatment with antibodies, bands were detected using an Immobilon Western System (Merck Millipore, WBKLS0100). For immunoprecipitation, cells plated on dishes were collected using PBS containing 2.5 mg/mL trypsin and 1 mM EDTA solution and suspended in 0.1% Triton X-100, 300 mM NaCl, 300 mM sucrose, 1 mM MgCl_2_, 1 mM EGTA, and 10 mM PIPES (pH 7.0) supplemented with 2% volume of protease inhibitor cocktail (Nacalai Tesque, 25955–24) (immunoprecipitation (IP) buffer). Cell lysates were immunoprecipitated using anti-Flag antibody-conjugated agarose beads (ANTI-FLAG M2 Agarose Affinity Gel, A4596, Sigma-Aldrich) and analyzed by Western blotting. Antibodies used for Western blotting are shown below (Table 2).

### 2.5. Methods for Tissue Processing and Immunostaining

Routinely processed formalin-fixed and paraffin-embedded specimens from 39 Japanese patients with breast cancer resected from 2011 to 2012 at Kurume University Hospital were used. The breast cancer tissues included papillotubular, solid-tubular, scirrhous, mucinous, and micropapillary carcinomas. Formalin-fixed, paraffin-embedded sections were immunostained with BenchMarkXT (Ventana Medical Systems, Tucson, AZ, USA). An anti-PHF8 antibody (Abcam ab84779) was applied at a dilution of 1:1000.

Two pathologists, S.O. and H.Y., who did not know the clinical status of each patient, independently evaluated and interpreted the results of immunostaining using the Allred scoring system [45,46]. This study was approved by the institutional ethics review board of Kurume University (approval no. 18082).

The expression (staining) levels of PHF8 were classified into the following four categories: tumors that showed PHF8 staining similar to that in the non-neoplastic epithelial area were classified as 3, tumors that showed intermediate PHF8 staining were classified as 2, tumors that showed weak PHF8 staining were classified as 1, and tumors containing no identifiable PHF8 were classified as 0. To estimate the rate of staining, the numbers of positive and negative tumor cells in a field were counted, and the ratio of positive to total cells was expressed as one of the following six categories: 0/100 as 0, 0–1/100 as 1, 1/100–1/10 as 2, 1/10–1/3 as 3, 1/3–2/3 as 4, and 2/3–1/1 as 5. The ratio of stained cells was combined with the strength of the PHF8 staining to produce the PHF8 score. The sections were also characterized by ER, PgR, and HER2 immunostaining, as described previously [46].

### 2.6. Metabolic Labeling Assay of Newly Synthesized RNA Using 5-Ethynyl Uridine (EU)

Detection of temporal RNA synthesis was performed using the Click-iT^®^ RNA Imaging Assay kit (Thermo Fisher Scientific, (Invitrogen), Waltham, MA, USA, Catalog #C10329) [24], basically according to the manufacturer’s instructions. In brief, after 293T cells were plated on glass coverslips and transfected with various plasmids using FuGENE6 transfection reagent (Promega, Madison, WI, USA) and cultured for 3 d. Then, 5-ethynyl uridine (EU) was added to the medium, and cells were further cultured for 45 min. Cells were fixed with methanol for 30 min at −20 °C, and biosynthetic incorporation of EU into newly transcribed RNA was detected by the Click-iT reaction buffer. After cells were treated with 1% skim milk in PBS, cells expressing the Flag-KDM2As were detected using an anti-Flag rabbit monoclonal antibody and a goat anti-rabbit IgG (H&L) conjugated with Cy3.

### 2.7. Detection of Phosphorylated Peptide by Phos-Tag

Phos-tag ligand was purchased from Wako Pure Chemical Industries (Osaka, Japan). Western blotting with Zn^2+^-Phos-tag SDS-PAGE was performed, as described previously [47], with a slight modification. Cells were lysed in RIPA buffer (50 mM Tris-HCl (pH 7.4), 0.15 M NaCl, 0.25% sodium deoxycholate, 1.0% NP-40, and 1 mM EDTA) containing 1 mM dithiothreitol, 10 μg/mL aprotinin, 10 μg/mL leupeptin, 1 mM sodium orthovanadate, 20 mM β-glycerophosphate, and 1 mM phenyl-methylsulfonyl fluoride. The proteins were separated with Zn^2+^-Phos-tag SDS-PAGE and transferred onto an Immobilon-P nylon membrane (Millipore, Billerica, MA, USA). The membrane was blocked with BlockAce (Dainippon Sumitomo Pharmaceutical, Osaka, Japan), incubated with primary antibodies (anti-GFP mouse monoclonal antibody, Santa Cruz Biotechnology, Inc. Dallas, TX, USA, sc-9996), and then incubated with a horseradish peroxidase-conjugated anti-mouse IgG secondary antibody. Signals were visualized using the electrogenerated chemiluminescence (ECL) system (Thermo Fisher Scientific) with Amersham Image 600 (Cytiva, Tokyo, Japan).

### 2.8. Production of Anti-Phosphorylated Ser 731

To produce an anti-phosphopeptide antibody that specifically recognizes phosphorylated Serine 731 (Ser731), the phosphopeptide CLQLIHDPVpSPRGMV was used as an antigen and affinity purified. The antibody did not recognize unphosphorylated peptide when tested using the above peptide and the peptide CLQLIHDPVSPRGMV by ELISA.

### 2.9. RNA Extraction and Quantitative Reverse Transcription–Polymerase Chain Reaction (qRT-PCR)

Isolation of total RNA from cells and cDNA synthesis were performed as described previously [8,24]. In brief, total RNA was isolated from cells, and single-strand cDNA was synthesized using random primers. The products were diluted to 150 μL with distilled water, and 2.5 μL of the resultant single-strand cDNA was used as the template for qRT-PCR, using a KAPA SYBR Fast qPCR kit (NIPPON Genetics, Tokyo, Japan) with a CSF Connect^TM^ Real-Time PCR Detection System (Bio-Rad Laboratories, Inc., Hercules, CA, USA), according to the manufacturer’s instructions [8]. The sets of PCR primers for amplification are shown below (PCR primers). To measure rRNA transcription, we amplified the sequence immediately after the transcription start site (1 –155) in the external transcribed spacer (ETS). The RNA in this region is destroyed soon after transcription, and the amounts of the RNA reflect rRNA transcription levels [8]. The values were normalized using the same amounts of control mRNA for RNA polymerase II subunit a (Polr2a), β-actin, and β2-microglobulin (B2M). The sets of PCR primers for amplification used are shown below (Table 3).

### 2.10. Statistical Analyses

Student’s *t*-tests were applied for comparisons of two groups with continuous data. Analyses of variance (ANOVA) were applied for comparisons between three groups.

## 3. Results

### 3.1. PHF8 Binds to KDM2A and Is Required for Reduction in rRNA Transcription by Mild Starvation

Identification of proteins interacting with KDM2A would be a key to unraveling the mechanism by which KDM2A regulates rRNA transcription during starvation. Because PHF8 was reported to accumulate in nucleoli, bind to rRNA genes, and regulate rRNA transcription [35,36], we tested whether KDM2A bound to PHF8. First, the levels of extraction of proteins by IP buffer (0.1% Triton X-100, 300 mM NaCl, 300 mM sucrose, 1 mM MgCl_2_, 1 mM EGTA, and 10 mM PIPES, pH 7.0) were investigated. Both KDM2A and PHF8 were almost completely extracted by IP buffer (Appendix A). The protein levels in the supernatant of IP buffer extraction were similar to those extracted by 3% SDS solution. No proteins (KDM2A and PHF8) were detected in the precipitated fractions of IP buffer extraction. The extraction mode of β-actin was similar to those of KDM2A and PHF8. However, histone H3 was mainly recovered in the precipitated fraction. These results indicate that KDM2A and PHF8 are well extracted in IP buffer. When Flag-tagged PHF8 (Flag-PHF8) was co-expressed with KDM2A in cells and immunoprecipitated by an anti-Flag antibody, KDM2A was detected in the fractions co-precipitated with Flag-PHF8 (Figure 1A). These results suggest that KDM2A interacts with PHF8.

Previously, we reported that KDM2A was expressed in breast cancer tissues [23]. To investigate the expression of PHF8 in breast cancer tissues, tumors from surgical breast cancer specimens, including papillotubular, solid-tubular, scirrhous, mucinous, and micropapillary carcinomas, were resected, and PHF8 was detected immunohistochemically. Staining of PHF8 was found in the nuclear regions of cells in both neoplastic and non-neoplastic areas (Appendix A). Staining indexes of PHF8 (PHF8 score) in 37 neoplastic areas were determined (Table 4). The PHF8 scores of all specimens ranged from 4 to 8. These results showed that PHF8 was expressed in breast carcinomas irrespective of the presence or absence of ER, PR, and/or overexpressed HER2 (Table 4). 

We previously showed that starvation by 2-deoxy-D-glucose (2DG) reduced the rRNA transcription and cell proliferation of breast cancer cells in a KDM2A–dependent manner [23,24]. The reductions in rRNA transcription and cell numbers were suppressed by either KDM2A or PHF8 knockdown (KD) in the breast cancer cell line MCF-7 (Figure 1B–D). We also tested whether PHF8 was involved in the reduction in rRNA transcription in the TNBC cell line MDA-MB-231 on 2DG treatment. The reductions in rRNA transcription and cell numbers were suppressed by either KDM2A or PHF8 KD in MDA-MB-231 cells (Appendix A). These results suggest that PHF8 and KDM2A are involved in the reduction in rRNA transcription and cell proliferation in breast cancer cells.

### 3.2. Identification of KDM2A Regions That Are Involved in Binding to PHF8

To detect the region(s) of KDM2A responsible for binding to PHF8, deletion mutants of KDM2A were expressed together with Flag-PHF8 in 293T cells. While wild-type KDM2A (amino acids 1–1162) was coprecipitated with Flag-PHF8 (Figure 2A), the mutant KDM2A (amino acids 1–824) that lacked the sequence on the C-terminal side was hardly coprecipitated with Flag-PHF8 (Figure 2A). Next, deletion mutants of KDM2A fused with green fluorescent protein (GFP) were expressed in 293T cells (Figure 2B). The GFP-fusion protein with the C-terminal half of KDM2A (amino acids 667–1162) was coprecipitated with Flag-PHF8, while that with the N-terminal portion of KDM2A (amino acids 1–732) was scarcely coprecipitated with Flag-PHF8 (Figure 2B). These results suggest that the C-terminal half of KDM2A (amino acids 667–1162) had an important segment for PHF8 binding. In the C-terminal region, there are several domains (Figure 2C). When the KDM2A mutant lacking a PHD domain, F-box domain, or LRR domain was tested for PHF8 binding, no domain was found to have affected the binding to PHF8 (Appendix A).

Next, a panel of deletion mutants of GFP fusion with KDM2A (amino acids 667–1162) were coexpressed with Flag-PHF8 in cells to detect the regions responsible for the binding with PHF8 (Appendix A). Finally, two regions of KDM2A (amino acids 822–946) and KDM2A (amino acids 1119–1162) were detected as the shortest PHF8–binding fragments (Figure 2C). Neither KDM2A (amino acids 822–898) nor KDM2A (amino acids 893–946) was sufficient to bind PHF8 (Appendix A and Figure 2C). After this, we designated the two regions, KDM2A (amino acids 822–946) and KDM2A (amino acids 1119–1162), as A and B regions for PHF8 binding, respectively (Figure 2C,E). When mutant KDM2As lacking an A or B region were coexpressed with Flag-PHF8, the mutants with either deleted region were found to reduce the binding of KDM2A to PHF8 (Figure 2D). These results confirmed that both A and B regions contributed to binding between KDM2A and PHF8. When intrinsically disordered regions (IDRs) were predicted using a published IDR-detecting program (https://iupred.elte.hu) [44], we detected five IDRs in KDM2A, and the A region contained part of IDR-4 (Figure 2E).

### 3.3. KDM2A with Deleted a Region Reduces rRNA Transcription More Strongly than Wild-Type KDM2A

IDRs play important roles in many cellular activities, complementing the functions of structured proteins and domains [27,28,29,30,31,32]. However, it is not clear which function each IDR plays. To investigate whether the A region was involved in controlling the activities of KDM2A to regulate rRNA transcription in cells, Flag-KDM2A or Flag-KDM2A with a deleted A region (Flag-KDM2A (dA region)) was expressed in 293T cells, and rRNA transcription was validated by a metabolic labeling assay [24], because 293T cells showed the same reduction in rRNA transcription by 2DG as MCF-7 cells did (Appendix A). As shown in Figure 3A, Flag-negative cells transfected with Flag-KDM2A (dA region) had similar levels of positive EU-signal cells as those with Flag-KDM2A (wild type). In contrast, Flag-positive cells with a Flag-KDM2A (dA region) had fewer EU signal–positive cells than those with Flag-KDM2A (wild type). These results were confirmed by counting cells with EU-positive cells in Flag-positive or -negative cells expressing Flag-KDM2A (wild) or Flag-KDM2A (dA region) (Figure 3B).

Next, rRNA transcription was detected biochemically. Cells were transfected with siRNA for KDM2A and cultured for 2 d, then transfected with various amounts of an expression vector encoding KDM2A or KDM2A (dA region) and cultured for two more days. Cell lysates were analyzed by Western blotting to detect KDM2As (Figure 3C), and it was found that KDM2A (dA region) was expressed at comparable levels to those of wild-type KDM2A, with increasing protein levels depending on plasmid amounts. Total RNA was isolated and analyzed by RT-PCR to detect pre-ribosomal RNA amounts. It was found that the levels of rRNA transcription in cells expressing KDM2A (dA region) were less compared to those in cells expressing KDM2A (Figure 3D).

The effect of deletion of the A region in KDM2A was also examined in breast cancer MCF-7 cells. After MCF-7 cells were transfected with an expression vector encoding KDM2A or KDM2A lacking the A region, cells were transfected with KDM2A siRNA (KDM2A), cultured for 2 d, and KDM2A protein and rRNA transcription were detected (Appendix A). While the level of KDM2A (dA region) was comparable to the wild type, cells expressing KDM2A (dA region) transcribed lower levels of pre-rRNA than cells expressing wild-type KDM2A. Together, these results suggest that KDM2A (dA region) more strongly reduces rRNA transcription than the wild-type KDM2A.

### 3.4. KDM2A Fragment That Contains the a Region Shows Switchable Interaction with PHF8

The A region contains IDR, and some IDRs show switchable protein–protein interactions [28]. We suspected that the stronger activity of KDM2A (dA region) to reduce rRNA transcription than wild-type KDM2A (Figure 3) may reflect the state in which binding of the A region to PHF8 was abolished. Thus, we hypothesized that binding of the A region to PHF8 suppressed the KDM2A activity to reduce rRNA transcription, and 2DG reduced the interaction between the A region and PHF8. To test this possibility, three GFP-fusion KDM2A fragments, GFP-KDM2A (amino acids 667–1162), GFP-KDM2A (amino acids 667–1118), which retained the A region but not the B region, and GFP-KDM2A (amino acids 667–1162, d822–946), which retained the B region but not the A region, were expressed with Flag-PHF8. Cells were treated with 2DG for 2 h, and proteins were collected using the anti-Flag antibody (Figure 4A). While the binding of GFP-KDM2A (amino acids 667–1162) and GFP-KDM2A (amino acids 667–1162, d822–946) to PHF8 was increased by 2DG, that of GFP-KDM2A (amino acids 667–1118) to PHF8 was reduced by 2DG (Figure 4A). These results suggest that binding between the A region and PHF8 was reduced by 2DG.

In our previous study, we reported that 2DG activated AMPK to induce the KDM2A–dependent reduction in rRNA transcription [23]. The reduction in rRNA transcription by AMPK activation by 5-aminoimidazole-4-carboxamide-1-β-D-ribofuranoside (AICAR) was confirmed (Figure 4B). Then, we searched the literature for amino acids whose phosphorylation status was affected by AMPK. Among the phosphorylated amino acids, phosphorylation of Ser731 of KDM2A, which was the nearest position to the A region, may be affected by AMPK [49]. To check the phosphorylation of the amino acid, a GFP-fusion protein containing amino acids 714–824 of KDM2A was examined by Phos-tag Western blotting [47]. The GFP-fusion protein expressed in 293T cells produced the shifted band, and the band was scarcely observed when Ser731 was mutated to Ala (Appendix A). These results suggest that Ser731 was phosphorylated in the cells.

Because Ser731 was not in the A region, the PHF8-binding region was re-evaluated. Flag-A region (amino acids 822–946), Flag-KDM2A (amino acids 667–946), which contained the entire IDR-4, or Flag-KDM2A (amino acids 667–898), which contained the entire IDR-4 but not the F-box domain, were co-expressed with PHF8 in 293T cells (Figure 2E and Figure 4C). The results showed that the binding of Flag-KDM2A (amino acids 667–946) to PHF8 was stronger than that of the Flag-A region (amino acids 822–946) (Figure 4C), indicating that the addition of KDM2A (amino acids 667–821), which was part of IDR-4 and contained Ser731, elevated the binding of the KDM2A (amino acids 822–946) to PHF8. Then, we named Flag-KDM2A (amino acids 667–946) the A’ region (Figure 2E and Figure 4C). Flag-KDM2A (amino acids 667–898) that lacked the F-box hardly bound PHF8 (Figure 4C).

Next, a specific antibody to phosphorylated Ser731 was produced. The binding specificity of the antibody to phosphorylated Ser731 was confirmed (Appendix A). When cells were transfected with an expression vector encoding KDM2A, the phosphorylation level of Ser731 was reduced by 2DG (Appendix A). 2DG, or the AMPK activator, also reduced the phosphorylation level of Ser731 of the endogenous KDM2A in MCF7 cells (Figure 4D).

### 3.5. A Dephosphorylation-Mimicked Mutation at Ser731 Reduced the Interaction Between PHF8 and the KDM2A Fragments and rRNA Transcription

To test whether dephosphorylation of Ser731 affected the interaction between PHF8 and the KDM2A fragments, the plasmid expressing the KDM2A fragments in which Ser731 was replaced with Ala (S731A), which resembles a dephosphorylation mimic mutation, was transfected to cells. When the GFP-A’ region (KDM2A amino acids 667–946) or GFP-A’ region (amino acids 667–946, S731A) was coexpressed with Flag-PHF8 in cells and collected by an anti-Flag antibody, the S731A mutation was found to have reduced the binding of GFP-KDM2A (amino acids 667–946) to PHF8 (Figure 5A). Similarly, when GFP-KDM2A (amino acids 667–1118) or GFP-KDM2A (amino acids 667–1118, S731A) was coexpressed with Flag-PHF8 in 293T cells and collected by anti-Flag antibody, it was also found that the binding of GFP-KDM2A (amino acids 667–1118) to PHF8 was reduced by an S731A mutation (Figure 5A). These results indicate that the S731A mutation specifically reduced the interaction between A regions in KDM2A and PHF8.

Next, the effect of the S731A mutation in KDM2A on rRNA transcription was investigated in MCF-7 cells. Cells were transfected with the expression vector encoding wild KDM2A or KDM2A (S731A), whose expression was resistant to siKDM2A, and then siRNA for endogenous KDM2A was transfected. The KDM2A protein levels in cells with the KDM2A mutant were similar to those in cells with the wild-type KDM2A (Figure 5). Total RNAs were isolated, and rRNA transcription was measured. The rRNA transcription in cells expressing KDM2A (S731A) was lower than that in cells with wild-type KDM2A (Figure 5A). These results suggest that dephosphorylation of Ser731 of KDM2A reduces rRNA transcription in cells. To test whether the S731A mutation of KDM2A affected cell proliferation, numbers of cells treated as in Figure 5A were counted. The numbers of cells expressing KDM2A (S731A) were lower than those in cells with the wild-type KDM2A (Figure 5C). These results suggest that the S731A mutation of KDM2A reduced cell proliferation. Together, our results presented here demonstrate that AMPK dephosphorylates Ser731 of KDM2A to reduce binding between the A region of KDM2A and PHF8 and reduces rRNA transcription (Figure 5D).

## 4. Discussion

Here, we found that PHF8 bound to KDM2A and was required for repression of rRNA transcription by 2DG. KDM2A bound to PHF8 through two regions. We named KDM2A (amino acids 822–946) and KDM2A (amino acids 1119–1162) the A and B regions, respectively (see Figure 2E). KDM2A with the A region deleted showed a stronger reduction in rRNA transcription. The binding of PHF8 to the KDM2A fragment (amino acids 667–1118) containing the A region but not the B region was reduced by 2DG. The KDM2A fragment (amino acids 667–946), which had the longer IDR, including Ser731, bound PHF8 more strongly than the A region. A 2DG or AMPK activator dephosphorylated Ser731. The replacement of Ser731 with Ala (S731A) of KDM2A, a dephosphorylation mimic mutation, reduced the binding between PHF8 and the KDM2A fragment (amino acids 667–946) or KDM2A fragment (amino acids 667–1118). The S731A mutation of KDM2A reduced rRNA transcription and cell proliferation. These results demonstrate that the PHF8-binding region of KDM2A played an important role in controlling KDM2A activity and suggest that a post-translational modification, dephosphorylation of Ser731 (loss of negative charge), affected the PHF8-binding property of the IDR-containing KDM2A fragments (amino acids 667–946 and 667–1118), resulting in a reduction in rRNA transcription. In this study, we connected the IDR of KDM2A to a specific molecular function in which KDM2A was regulated via AMPK through dephosphorylation of a specific Ser in IDR. Our results provide a novel mechanism by which epigenetic protein KDM2A is regulated by another epigenetic protein, PHF8, through AMPK. The conservation of amino acid sequences in proteins between species or within family proteins provides clues as to whether the sequences are important to the functions. When the amino acid sequence around Ser731 of human KDM2A (amino acid 700–808 in IDR-4) was compared to those of mouse and chicken, the clear conservation was identified between the species (99% identical between human and mouse and 91.7% identical between human and chicken). The identical levels in sequences containing Ser731 are similar to those of the sequences of the JmjC domain (Table 5). These results highlight that this region has an important function and support the importance of the regulatory mechanism identified here. In contrast, this conserved serine is absent in KDM2B (NP_115979.3), a KDM2A paralog. The lack of the conserved serine would suggest that KDM2B does not respond to starvation as KDM2A does.

The amino acid sequence for human (*Homo sapiens*) KDM2A was taken from NP_036440.1, that for mouse (*Mus musculus*) KDM2A was taken from NP_001001984.2, and that for chicken (*Gallus gallus*) KDM2A was taken from XP_015128112.1.

The mechanism by which the dissociation between the PHF8-A region of KDM2A allows rRNA suppression is an important question. KDM2A binds to PHF8 through the B region, in addition to the A region, and the wild-type KDM2A fragment and the fragment retaining the B region remain bound to PHF8 after 2DG treatment (Figure 3A). We previously showed that KDM2A binds to the rRNA gene promoter through the CXXC-ZF domain [9], and the amount of KDM2A bound to the rDNA promoter does not change after starvation [8,23]. While PHF8 is recruited to the rRNA gene by binding to the active histone marks H3K4me3, the level of H3K4me3 remained unchanged during starvation [8]. These results suggest that the two proteins continuously remain present on the rRNA gene promoter after 2DG treatment, but the binding mode between PHF8 and KDM2A may be altered by starvation. It is possible that the A region released from PHF8 on starvation binds another molecule and reduces rRNA transcription. This putative binding protein might contain a component of repressive complexes. Alternatively, the dissociation may affect the histone demethylase activities of KDM2A and PHF8. Further studies are needed to clarify these points.

In mammals, mTORC1 phosphorylates some transcription factors, including TIF-IA, for the formation of rRNA polymerase I (Pol I) transcription initiation complex [50,51,52]. AMPK phosphorylates TSC2, resulting in inhibition of mTORC1 [52]. TIF-IA binding to the rRNA gene promoter is reduced, and rRNA transcription is decreased [23,53]. Therefore, one way to reduce rRNA transcription by AMPK is that AMPK signaling reduces Pol I activity through TSC2-mediated TIF1A inhibition. Although treatment of cells with mild starvation decreased rRNA transcription in a KDM2A-dependent manner, this treatment did not decrease the levels of TIF-IA in the rRNA gene promoter [23]. These results suggest that KDM2A-mediated reduction in rRNA transcription does not require TIF-1A regulation by AMPK. So far, the molecular mechanisms by which KDM2A is dephosphorylated and how dephosphorylated KDM2A reduces rRNA transcription are not clear, and the cross talk between the AMPK downstream effectors has not been identified.

Here, we showed that KDM2A together with PHF8 controlled rRNA transcription in breast cancer cells. Both KDM2A and PHF8 are expressed in breast cancer tissues (Table 4) [23,24]. Dephosphorylation of Ser731 in KDM2A was induced by 2DG or AMPK activation, which activated KDM2A to reduce rRNA transcription. Because AMPK is a kinase, it is plausible that the kinase activity of AMPK activates a phosphatase or inactivates a kinase to reduce phosphorylation of Ser731. A precedent that AMPK activates a phosphatase, in which adiponectin-activated AMPK stimulates dephosphorylation of AKT through protein phosphatase 2A activation, has been reported [54]. When KDM2A was expressed in 293T cells, phosphorylated Ser731 was detected (Appendix A). Addition of 2DG or AMPK activator AICAR reduced the level of phosphorylated Ser731, and the potent and selective inhibitor of protein phosphatase 2A (PP2A), the phosphatase inhibitor cantharidin, prevented the reduction in pSer731 levels (Appendix A). These results suggest that PP2A is activated downstream of the AMPK pathway and dephosphorylates KDM2A at Ser731. It is also not known which enzyme phosphorylates Ser731. Identification of responsible phosphatase(s) and kinase(s) that control the phosphorylation status of Ser731 of KDM2A would open an avenue for a novel target of breast cancer therapy.

## 5. Conclusions

PHF8 bound KDM2A through two regions of KDM2A. One of the regions contained an IDR. The IDR may be involved in rapidly switchable protein–protein interactions. An AMPK activator dephosphorylated KDM2A at Ser731, which was located in the IDR overlapping with the PHF8-binding region. The mode of interaction between KDM2A and PHF8 is regulated via dephosphorylation of Ser731 of KDM2A through AMPK to control rRNA transcription. These findings provide a clue to how the epigenetic protein KDM2A is regulated. Identification of the enzyme(s) that control the phosphorylation state of Ser731 of KDM2A would provide a novel target for breast cancer therapy.

## Figures and Tables

**Figure 1 biomolecules-15-00661-f001:**
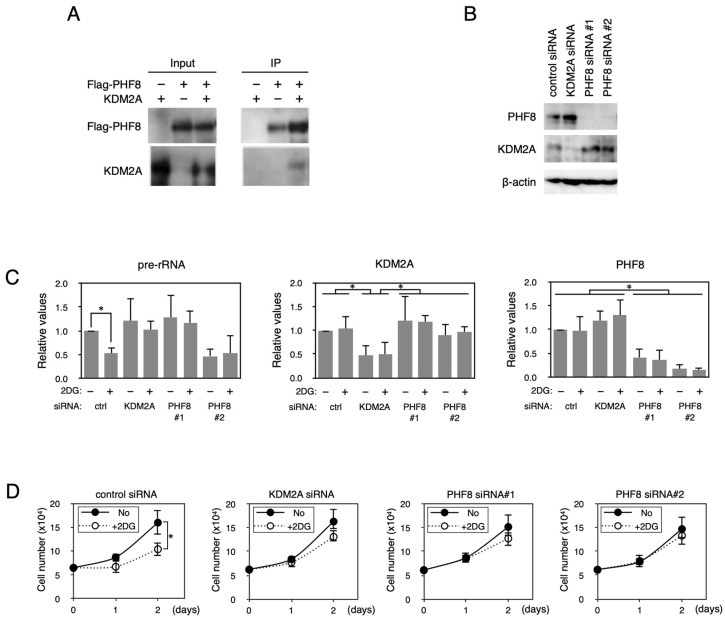
PHF8 binds to KDM2A and is involved in the control of rRNA transcription. (**A**) Binding of KDM2A to PHF8. An expression vector encoding Flag-PHF8 or the empty vector was cotransfected with an expression vector encoding KDM2A or an empty vector into 293T cells by FuGENE6. Cell lysates were immunoprecipitated with anti-Flag antibody–conjugated agarose and analyzed by Western blotting with an anti-KDM2A antibody (Proteintech, Rosemont, IL, USA) and a rabbit anti-Flag antibody (Sigma). One-tenth of the input samples were also analyzed. Detection for immunoprecipitated KDM2A was enhanced by long exposure compared to the input sample. (**B**) Knockdown of *PHF8* and *KDM2A*. MCF-7 cells were transfected with siRNAs for *KDM2A* or *PHF8* (siPHF8#1 and siPHF8#2) for 3 d, and cell lysates were analyzed by Western blotting using anti-PHF8 (Cell Signaling Technology, Danvers, MA, USA) and anti-KDM2A (Proteintech) antibodies to detect PHF8 and KDM2A, respectively. β-actin was detected for a loading control. (**C**) Involvement of PHF8 in reduction in rRNA transcription by 2DG. Cells transfected as in (**B**) were replated in the growth medium. The next day, cells were cultured in the presence or absence of 2 mM 2DG for 2 h, total RNA was isolated, and the pre-rRNA, KDM2A mRNA, and PHF8 mRNA were detected by RT-PCR. The results were normalized by the values of Polr2a mRNA. (**D**) Involvement of PHF8 in 2DG–induced reduction in cell proliferation. Cells transfected as in (**B**) were replated and cultured in medium in the presence or absence of 2 mM 2DG. On the indicated days, cell numbers were counted. All experiments were performed three times, and mean values with the standard deviations are indicated. *, *p* < 0.05.

**Figure 2 biomolecules-15-00661-f002:**
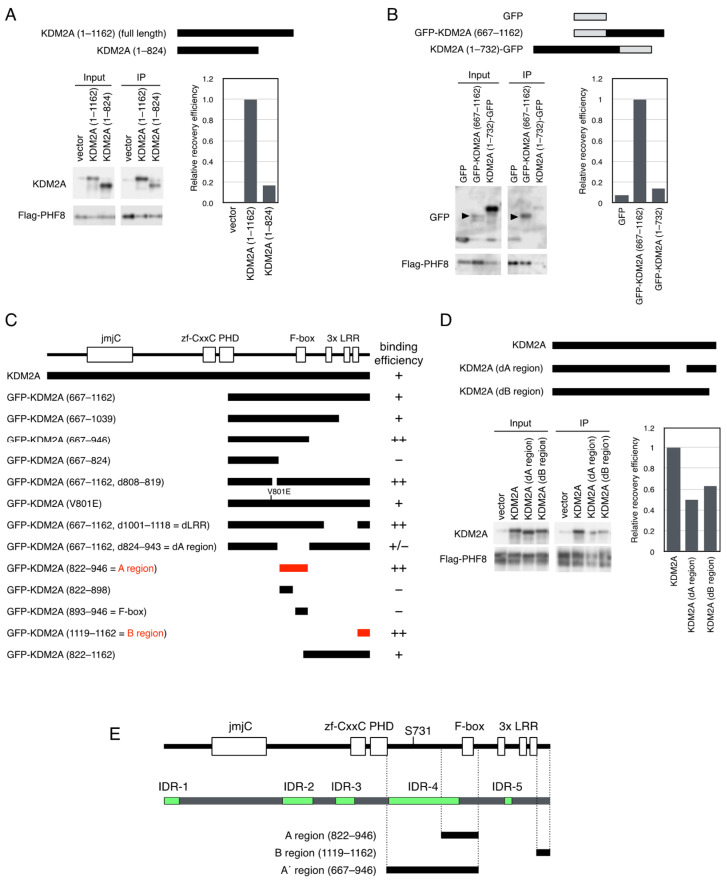
Identification of the regions of KDM2A involved in binding to PHF8. (**A**) An expression vector encoding KDM2A (full length, amino acids 1–1162) and KDM2A (amino acids 1–824) was cotransfected with a Flag-PHF8 expression vector into 293T cells by electroporation. Cell lysates were immunoprecipitated with anti-Flag antibody–conjugated agarose and analyzed by Western blotting with anti-KDM2A antibody (Proteintech) and rabbit anti-Flag antibody (left-hand lower panel). Then, 0.28% of input samples for KDM2A and 16% of input samples for Flag-PHF8 were also analyzed. The relative recovery efficiency, KDM2A (IP/input)/PHF8 (IP), was expressed as the signals of immunoprecipitated KDM2A against input KDM2A signals, which were divided by the values of immunoprecipitated PHF8 (right-hand lower panel). Diagrams of the expressed proteins are shown (upper panel). (**B**) An expression vector encoding GFP protein fused with KDM2A fragments, GFP-KDM2A (amino acids 667–1162) and KDM2A (amino acids 1–732)-GFP, was cotransfected with an expression vector encoding Flag-PHF8 in 293T cells by electroporation. Cell lysates were collected and analyzed as described in (**A**) using anti-GFP antibody (Abcam) and anti-PHF8 antibody (Cell Signaling Technology) (left-hand lower panel). Arrowheads show the positions of GFP-KDM2A (667–1162). Moreover, 0.28% of input samples for GFP and 16% of input samples for Flag-PHF8 were analyzed. The relative recovery efficiency, GFP (IP/input)/PHF8 (IP), was expressed as the signals of immunoprecipitated GFP against input GFP signals, which were divided by the values of immunoprecipitated PHF8 (right-hand lower panel). The diagrams of the expressed proteins were shown (upper panel). (**C**) Summary of the binding of KDM2A fragments to PHF8. The names of GFP proteins fused with the KDM2A fragments are shown. At the top, functional folded domains of KDM2A are shown. On the right-hand side, the binding efficiencies are expressed as follows: −: negative; +/−: weak; +: moderate; ++: strong. KDM2A (amino acids 822–946) and KDM2A (amino acids 1119–1162) were identified as the shortest fragments binding to PHF8 and designated as the (**A**,**B**) regions, respectively. (**D**) An expression vector encoding KDM2A, KDM2A lacking an A region, shown as KDM2A (dA region), or KDM2A lacking a B region, shown as KDM2A (dB region), was cotransfected with an expression vector encoding Flag-PHF8 in 293T cells by electroporation. Cell lysates were collected and analyzed as in (**A**) by Western blotting with an anti-KDM2A antibody (Proteintech) and an anti-PHF8 antibody (Cell Signaling Technology) (left-hand lower panel). Input samples for KDM2A (0.28%) and input samples for Flag-PHF8 (16%) were also analyzed. The relative recovery efficiency, KDM2A(IP/input)/PHF8 (IP), was expressed as the signals of immunoprecipitated KDM2A against input KDM2A signals, which were divided by the values of immunoprecipitated PHF8 (right-hand lower panel). Diagrams of the expressed proteins are shown (upper panel). (**E**) The structure of human KDM2A protein. Human KDM2A proteins with functional folded domains (upper panel), IDRs (middle panel), and functional regions identified in this study (lower panel) are shown. The functional domains were detected by the SMART program [48]. IDRs were predicted using a database of disordered protein predictions [44]. The regions involved in KDM2A functions identified here are the A region (KDM2A amino acids 822–946), B region (KDM2A amino acids 1119–1162), and A’ region.

**Figure 3 biomolecules-15-00661-f003:**
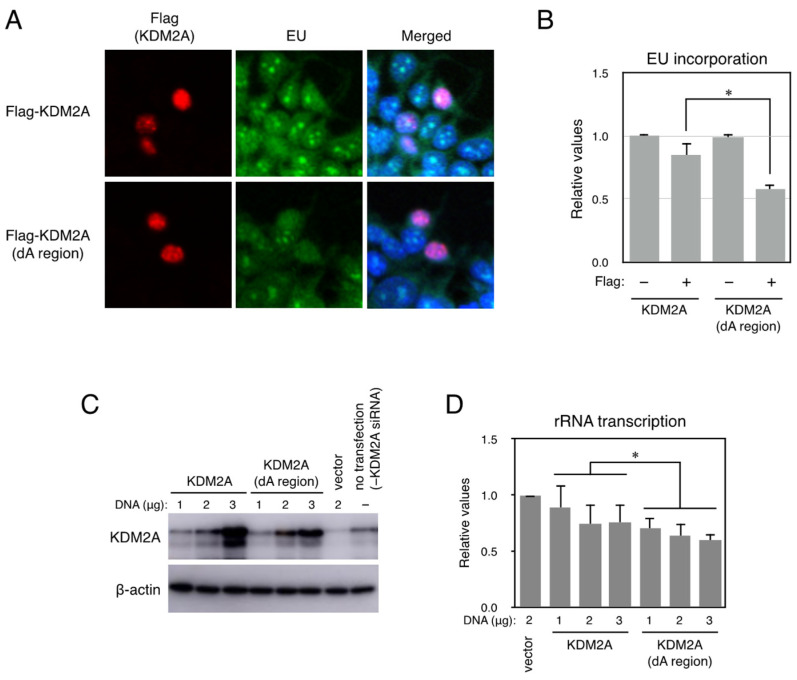
Cells expressing the KDM2A-deleted A region transcribed lower levels of pre-rRNA than cells expressing wild KDM2A. (**A**) 293T cells were transfected with plasmids encoding Flag-KDM2A or Flag-KDM2A (dA region) by FuGENE6. After 3 d, cells were cultured in the presence of EU for 45 min. Cells were fixed with methanol for 20 min at −20 °C, and EU incorporated into cells was detected. After that Flag, signals were detected using an anti-Flag rabbit monoclonal antibody (Cell Signaling Technology) and anti-rabbit IgG conjugated with Cy3 reagent. Cells were observed under fluorescent microscopy. (**B**) After taking pictures of cells in (**A**), numbers of EU-positive cells in each condition were counted, and the rates of EU-positive cell numbers against total cell numbers were expressed. All experiments were performed three times, and mean values with the standard deviations are indicated. *, *p* < 0.05. Scale bars show 10 μm. (**C**) The levels of rRNA transcription in cells expressing the KDM2A-deleted A region were lower than those in cells expressing wild-type KDM2A. After 293T cells were transfected with KDM2A siRNA (KDM2A) and cultured for two days, cells were transfected with various amounts of expression vector encoding KDM2A or KDM2A lacking the A region (KDM2A dA region), whose expression was resistant to siKDM2A [9,24], by electroporation. Cell lysates were analyzed by Western blotting using an anti-KDM2A antibody (Abcam, ab191387). β-actin was detected as a loading control. (**D**) Total RNA was isolated from cells prepared, and the levels of rRNA transcription were detected by RT-PCR. The results were normalized by the values of β-actin mRNA. The experiment was performed three times, and mean values with the standard deviations are indicated. *, *p* < 0.05.

**Figure 4 biomolecules-15-00661-f004:**
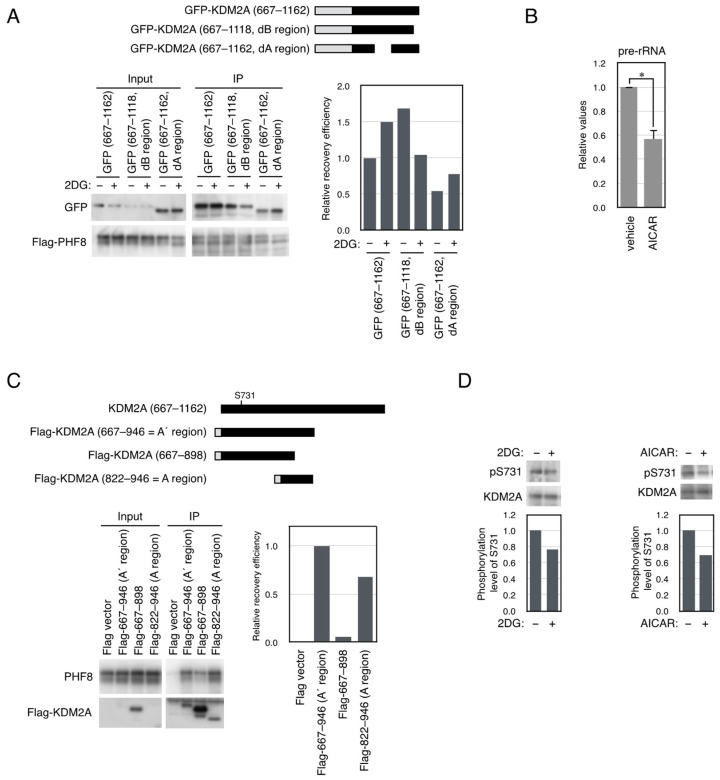
Effect of 2DG on binding properties of KDM2A fragments to PHF8 and phosphorylation at Ser731 of KDM2A. (**A**) 2DG reduced the binding between PHF8 and the KDM2A fragment containing an A region but not a B region. An expression vector encoding GFP-KDM2A (amino acids 667–1162), shown as GFP-wild, GFP-KDM2A (amino acids 667–1162 dB region), shown as the GFP-dB region, or GFP-KDM2A (amino acids 667–1162 dA region), shown as the GFP-dA region, was cotransfected with an expression vector encoding Flag-PHF8 in 293T cells by electroporation. Cells were cultured for 2 d and treated with or without 2 mM 2DG for 2 h. Cell lysates were collected and immunoprecipitated by an anti-Flag antibody-conjugated agarose and analyzed by Western blotting using an anti-GFP antibody (Abcam) and an anti-PHF8 antibody (Cell Signaling Technology) (left-hand lower panel); 0.3% of input samples for GFP and 12.5% of input samples for Flag-PHF8 were also analyzed. The relative recovery efficiency, GFP (IP/input)/PHF8 (IP), was expressed as the signals of immunoprecipitated GFP against input GFP signals, which were divided by the values of immunoprecipitated PHF8 (right-hand lower panel). Diagrams of the expressed proteins are shown (upper panel). (**B**) AMPK activation reduced rRNA transcription. MCF-7 cells were replated in the growth medium and were cultured in the presence or absence of 0.5 mM AICAR for 9 h, and total RNA was isolated and the pre-rRNA was detected by RT-PCR. The results were normalized by the values of β-actin mRNA. The experiments were performed four times, and mean values with the standard deviations are indicated. *, *p* < 0.05. (**C**) The A’ region that contains the A region with the entire ID-4 region binds more strongly to PHF8 than the A region alone. An expression vector encoding Flag-tagged fragments of KDM2A was co-transfected with a PHF8 expression vector into 293T cells by electroporation. Flag-tagged proteins were Flag-KDM2A (amino acids 667–946), shown as Flag-667-946 (A’ region); Flag-KDM2A (amino acids 667–898), shown as Flag-667–898; and Flag-KDM2A (amino acids 822–946), shown as Flag-822–946 (A region) (left-hand upper panel). Cell lysates were immunoprecipitated with anti-Flag antibody-conjugated agarose and analyzed by Western blotting with anti-GFP antibody and rabbit anti-Flag antibody (Sigma) (right-hand upper panel), and 0.7% of input samples were also analyzed for PHF8. One-third of input samples were also analyzed for KDM2A (FbxL11). The relative recovery efficiency, PHF8 (IP/input)/FlagKDM2A (IP), was expressed as the signals of immunoprecipitated PHF8 against the input PHF8 signals, which were divided by immunoprecipitated Flag-KDM2A (lower panel). (**D**) The phosphorylation level of Ser731 of endogenous KDM2A in breast cancer MCF-7 cells was reduced by 2DG or an AMPK activator. MCF7 cells were treated with 2DG or AICAR for 2 h and 3 h, respectively, and cell lysates were analyzed by Western blotting using antibodies to the phosphorylated Ser731 (shown as pS731) and KDM2A (Abcam, ab191387) (upper panels). The phosphorylation levels were expressed as the signals of phosphorylated Ser731 divided by those of KDM2A (lower panels).

**Figure 5 biomolecules-15-00661-f005:**
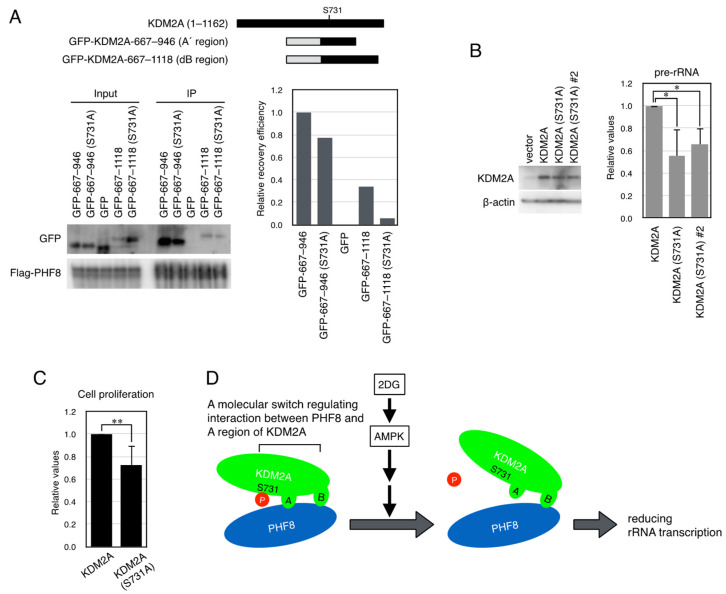
KDM2A with the starvation-mimicking mutation S731A reduced the interaction between the A region and PHF8 and rRNA transcription. (**A**) An expression vector encoding the GFP-A’ region (KDM2A amino acids 667–946) shown as GFP-667–946, the GFP-A’ region (KDM2A amino acids 667–946, S731A) shown as GFP-667–946 (S731A), GFP, GFP-KDM2A (amino acids 667–11118) shown as GFP-667–1118, or GFP-KDM2A (amino acids 667–1118, S731A) shown as GFP-667-1118 (S731A) was cotransfected with an expression vector encoding Flag-PHF8 in 293T cells by electroporation (upper panel). Cell lysates were collected and immunoprecipitated by anti-Flag antibody-conjugated agarose and analyzed by Western blotting using an anti-GFP antibody and an anti-PHF8 antibody (Cell Signaling Technology) (left-hand lower panel), and 1.1% of input samples for GFP and one-third of input samples for Flag-PHF8 were also analyzed. The relative recovery efficiency, GFP (IP/input)/PHF8 (IP), was expressed as the signals of immunoprecipitated GFP against input GFP signals, which were divided by immunoprecipitated PHF8 (right-hand lower panel). A diagram of the expressed proteins is shown (upper panel). (**B**) The S731A mutation elevated KDM2A activity to reduce rRNA transcription in MCF-7 cells. After MCF-7 cells were transfected with an expression vector encoding KDM2A, shown as KDM2A, or the S731A-mutant KDM2A, shown as KDM2A (S731A) or S731KDM2A (S731A)#2 (60% of the plasmid amount compared to that of wild-type KDM2A) by electroporation and then KDM2A siRNA by RNAiMAX, cells were cultured for 2 d. The *KDM2A* cDNAs, whose expression was resistant to siKDM2A [9,24], were used here. Cell lysates were analyzed by Western blotting using anti-KDM2A antibody (Abcam, ab191387) to detect KDM2A. β-actin was detected as a loading control (left-hand panel). RNA was isolated, and the levels of rRNA transcription were detected by RT-PCR. The results were normalized by the amount of B2M (right-hand panel). The experiments were performed four times, and mean values with the standard deviations are indicated. *, *p* < 0.05. (**C**) The point mutation of S731A in KDM2A reduced cell proliferation in MCF-7 cells. After MCF-7 cells were transfected as described in (**B**), the numbers of cells were counted, and relative numbers of cells transfected with KDM2A (S731A) against cells transfected with wild-type KDM2A are shown. The experiments were repeated seven times, and mean values with the standard deviations are indicated. **, *p* < 0.005. (**D**) Model of the control of rRNA transcription by KDM2A by starvation. Glycolysis inhibitor 2DG or AICAR activated AMPK, which dephosphorylated Ser731 of KDM2A to reduce interaction between the A region and PHF8, and then reduced rRNA transcription.

**Table 1 biomolecules-15-00661-t001:** siRNA oligonucleotide sequences.

Stealth siRNA	siRNA Sequence
siKDM2A	5′-GAACCCGAAGAAGAAAGGAUUCGUU-3′
siPHF8#1	5′-CAACAAAUGCCAAUCUGACUCUCUU-3′
siPHF8#2	5′-GAGCUCCGGAGUAGGACUUUUGACA-3′
control siRNA	Stealth RNAi Negative Control Medium GC Duplex, Thermo Fisher

**Table 2 biomolecules-15-00661-t002:** Antibodies.

Antigen	Antibody
KDM2A	Rabbit anti-KDM2A polyclonal antibody (Proteintech, 24311-1-AP) Rabbit anti-KDM2A monoclonal antibody (Abcam, ab191387) Anti-FbxL11 (KDM2A) antibody (Abcam, ab99242)
PHF8	Rabbit anti-PHF8 monoclonal antibody (Cell Signaling Technology, PHF8 (E6K3Y))
Flag	Rabbit polyclonal anti-Flag antibody (Sigma, F7425) Mouse monoclonal anti-Flag antibody (M2, Sigma, F1804)
GFP	Rabbit polyclonal anti-GFP antibody (Abcam, ab290) Mouse monoclonal anti-GFP antibody (Santa Cruz, ab290)
β-actin	Mouse monoclonal anti-β-actin antibody (Sigma, AC-15)
Rabbit IgG	Goat-rabbit IgG-horseradish peroxidase (HRP) (Abcam, ab6721) Goat anti-rabbit IgG-HRP (Santa Cruz, sc-2054, human and mouse IgG adsorbed) Goat anti-rabbit IgG(H&L)-Cy3 conjugated (Zymed, #81-6115)
Mouse IgG	Goat anti-mouse IgG-HRP (Santa Cruz, sc-2005)

**Table 3 biomolecules-15-00661-t003:** PCR primers.

Gene	Primer Sequence
pre-rRNA	Forward: 5′-GCTGACACGCTGTCCTCTG-3′ Reverse: 5′-TCGGACGCGCGAGAGAAC-3′
KDM2A	Forward: 5′-TCCCCACACACATTTTGACATC-3′ Reverse: 5′-GGGGTGGCTTGAGAGATCCT-3′
PHF8	Forward: 5′-AGCCCTACGTTCGTCAGAGA-3′ Reverse: 5′-CAACCCATCCTTCTTCAGGA-3′
Polr2a	Forward: 5′-ATCTCTCCTGCCATGACACC-3′ Reverse: 5′-AGACCAGGCAGGGGAGTAAC-3′
β-actin	Forward: 5′-CGTCTTCCCCTCCATCGT-3′ Reverse: 5′-GAAGGTGTGGTGCCAGATTT-3
B2M	Forward: 5′-CTCGCGCTACTCTCTCTTTCT-3′ Reverse: 5′-TGTCGGATTGATGAAACCCAG-3′

**Table 4 biomolecules-15-00661-t004:** Expression of PHF8 protein in breast cancer tissues.

PHF8 score		**Papillotubular**					**Solid tubular**					**Scirrhpous**					**Mucinous**					**Miceopapillary**				
		4	5	6	7	8	4	5	6	7	8	4	5	6	7	8	4	5	6	7	8	4	5	6	7	8
Number		2	2	2	8	3	0	1	1	0	0	1	1	4	5	3	0	0	2	0	0	0	1	0	1	0
HER2	0	0	0	0	6	2	0	0	0	0	0	0	1	0	4	2	0	0	1	0	0	0	0	0	0	0
	1	1	0	0	1	1	0	0	0	0	0	0	0	1	0	0	0	0	1	0	0	0	0	0	0	0
	2	1	0	0	1	0	0	1	1	0	0	0	0	1	1	1	0	0	0	0	0	0	0	0	3	0
	3	0	2	2	0	0	0	0	0	0	0	1	0	2	0	0	0	0	0	0	0	0	1	0	0	0
ER	−	0	1	1	2	0	0	0	0	0	0	1	1	0	1	0	0	0	0	0	0	0	1	0	0	0
	+	2	1	1	6	3	0	1	1	0	0	0	0	4	4	3	0	0	2	0	0	0	0	0	1	0
PgR	−	0	2	2	2	0	0	1	0	0	0	1	0	1	1	0	0	0	1	0	0	0	1	0	0	0
	+	2	0	0	6	3	0	0	1	0	0	0	1	3	4	3	0	0	1	0	0	0	0	0	1	0
Triple Negative		0	0	0	2	0	0	0	0	0	0	0	0	0	0	0	0	0	0	0	0	0	0	0	0	0

PHF8 score, 0: negative; 1: weak; 2: intermediate; 3: strong.

**Table 5 biomolecules-15-00661-t005:** Identity to human KDM2A (amino acids 1162) amino acid sequence (%).

Species	Total AA	JmjC Domain	A Region	IDR-4	IDR-4 (AA 700-808)
Mouse	AA 1161 (97.3%)	168/169 (99.4%)	118/125 (94.4%)	208/217 (95.8%)	108/109 (99%)
Chicken	AA 1168 (86%)	156/169 (92%)	95/125 (76%)	171/217 (78.8%)	100/108 (91.7%)

## Data Availability

The original contributions presented in this study are included in the article. Further inquiries can be directed to the corresponding author(s).

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
