# Peer review of "Interaction Between PHF8 and a Segment of KDM2A, Which Is Controlled by the Phosphorylation Status at a Specific Serine in an Intrinsically Disordered Region of KDM2A, Regulates rRNA Transcription and Cell Proliferation in a Breast Cancer Cell Line"

_biomolecules, 2025, doi:10.3390/biom15050661_

Round 1

Reviewer 1 Report

Comments and Suggestions for Authors

The study elegantly connects metabolic stress (via 2-DG) to epigenetic regulation of rRNA through AMPK-mediated KDM2A-PHF8 interaction, revealing a phosphorylation-dependent switch at Ser731 as a critical regulatory node. This work significantly advances our understanding of metabolic-epigenetic crosstalk in cancer, revealing phosphorylation-regulated KDM2A-PHF8 interaction as a rheostat for rRNA control. With additional validation of the AMPK-KDM2A axis and in vivo studies, these findings could open new avenues for targeting ribosome biogenesis in breast cancer. But I have several following concerns:

  1. Direct evidence showing AMPK phosphorylates/dephosphorylates KDM2A at Ser731 (e.g., in vitro kinase assays with AMPK and KDM2A peptides) would strengthen the central claim
  2. Include rescue experiments (e.g., AMPK knockdown + phospho-mimetic Ser731 mutants) to confirm the AMPK-dependence of Ser731 modification.

  3. Test whether this mechanism is conserved in other breast cancer subtypes (e.g., TNBC) or glucose-starvation models (e.g., low-glucose media).

  4. Assess if Ser731Ala mutation affects tumor growth in vivo (xenograft models), which would bolster translational relevance.

  5. Explore how PHF8-KDM2A dissociation enables rRNA suppression: Does freed KDM2A recruit repressive complexes to rDNA loci? ChIP-seq of KDM2A/PHF8 at rRNA genes under 2-DG treatment could elucidate this.

  6. Determine if other AMPK downstream effectors (e.g., TSC2) cross-talk with this pathway.

  7. Discuss if Ser731 is conserved across species or KDM2A paralogs (KDM2B).

  8. Please use the standard three-line form for all the tales in the manuscript.
  9. Gene or nucleotide sequence names should be italicized in the manuscript.
  10. Please add a scale bar to the microscope picture taken in the text.
  11. Abbreviations should be defined when they first appear in the text. Such as "PHF8" in Line 24, "SDS-PAGE" in Line 122, ...please double check all the text and revsie them.
  12. Please unify the format of references in the article, including the author's name, the case of words in the title of the article, the writing of the name of the journal, and the page number.

Comments on the Quality of English Language

The English could be improved to more clearly express the research.

Author Response

Reviewer 1

Attached please find our revised manuscript entitled “Interaction between PHF8 and a segment of KDM2A, which is controlled by the phosphorylation status at a specific serine in an intrinsically disordered region of KDM2A, regulates rRNA transcription and cell proliferation in a breast cancer cell line”, which we are re-submitting to your consideration for publication in Biomolecules as an article. The manuscript was revised following the useful comments we received from the reviewers. We could respond to all the comments. We added the results of the new experiments in the revised version of the manuscript. The newly added figures are Figure S1, Figure S3, Figure S9, and the newly added table is table 2. The improved figures are Figure 3A, in which we added scale bars. Our conclusion could be far clearly stated in this new version.

 Point-by-point response to Comments (Referee #1)

Comments 1: Direct evidence showing AMPK phosphorylates/dephosphorylates KDM2A at Ser731 (e.g., in vitro kinase assays with AMPK and KDM2A peptides) would strengthen the central claim

Response 1: Thank you for your suggestion. The relationship between AMPK and regulation of the phosphorylation state of Ser731 in KDM2A is an important issue to address. Phosphorylated Ser731 was detected when KDM2A was expressed in 293T cells. Phosphorylated Ser731 was decreased by the addition of 2DG or the AMPK activator AICAR, and the decrease was prevented by the addition of cantharidin, a phosphatase inhibitor that strongly and selectively inhibits protein phosphatase 2A (PP2A). These results suggest that PP2A is activated by AMPK, which dephosphorylates KDM2A at Ser731. To illustrate these results, a new figure has been added as Figure S9 and is discussed in line 609, in discussion section as below.

Text:

When KDM2A was expressed in 293T cells, phosphorylated Ser731 was detected (Figure S9). Addition of 2DG or AMPK activator AICAR reduced the level of phosphorylated Ser731, and the potent and selective inhibitor of protein phosphatase 2A (PP2A), the phosphatase inhibitor cantharidin, prevented the reduction of pSer731 levels (Figure S9). These results suggest that PP2A is activated downstream of the AMPK pathway, and dephosphorylates KDM2A at Ser731.  

Figure S9. Evidence for phosphatase involvement in dephosphorylation of Ser731 of KDM2A on starvation. An expression vector encoding KDM2A was transfected into 293T cells by electroporation, and then siRNA for KDM2A was transfected. After cultured for 2 d, cells were treated with a phosphatase inhibitor, cantharidin, at the indicated concentrations, for 4 h in the presence or absence of 2mM 2DG (A) or 0.5 mM AICAR (B). Cells were collected and analyzed by Western blotting using anti-phosphorylated Ser731 antibody and anti-KDM2A antibody (Abcam, ab191387) (upper panel). The relative phosphorylation levels were expressed as the values of phosphorylated Ser731 divided by the values for KDM2A. The value without 2DG (A) or AICAR (B), and cantharidin was expressed as 1 (lower panel).

Comments 2: Include rescue experiments (e.g., AMPK knockdown + phospho-mimetic Ser731 mutants) to confirm the AMPK-dependence of Ser731 modification.

Response 2: As indicated in response to Question #1, experiments with the PP2A inhibitor cantharidin suggested that AMPK activates PP2A to reduce pS731 levels, supporting an AMPK-dependent nature of Ser731 modification. To illustrate these results, a new figure has been added as Figure S9 and is explained in line 609 of the Discussion section, as answered in response to Reviewer #1's Question 1.

Comments 3: Test whether this mechanism is conserved in other breast cancer subtypes (e.g., TNBC) or glucose-starvation models (e.g., low-glucose media).

Response 3: Thank you for pointing this out. We tested whether PHF8 is involved in the reduction of rRNA transcription by 2DG treatment in the TNBC cell line, MDA-MB-231. The results showed the involvement of PHF8 in response to starvation in TNBC cell line as well. The results were described in line 257 of text, and a new figure is added as Figure S3.  

Text: We also tested whether PHF8 was involved in reduction of rRNA transcription in TNBC cell line MDA-MB-231 on 2DG treatment. The reductions of rRNA transcription and cell numbers were suppressed by either KDM2A or PHF8 KD in MDA-MB-231 cells (Figure S3). These results suggest that PHF8 and KDM2A are involved in the reduction of rRNA transcription and cell proliferation in breast cancer cells.

Figure S3. Effects of PHF8 and KDM2A knockdown on TNBC cell line MDA-MB-231. (A) MDA-MB-231 cells (European collection of authenticated cell cultures, ECACC, Catalogue No. 92020424) were transfected with stealth siRNAs for KDM2A or PHF8 (siPHF8#1 and siPHF8#3), and cultured for 3 d. The oligonucleotide sequence of stealth siRNA siPHF8#3 is 5’-CAUUCCACUUCAGUGUCCAUGUCCA-3’. Cells were replated in the growth medium. The next day, cells were cultured in the presence or absence of 3 mM 2DG for 2 h. Total RNA was isolated and pre-rRNA, KDM2A mRNA, and PHF8 mRNA were detected by RT-PCR. Results were normalized by Polr2a mRNA value. (B) Involvement of KDM2A and PHF8 in inhibition of cell proliferation by 2DG. Cells transfected as in (A) were replated and cultured in medium in the presence or absence of 3 mM 2DG. On the indicated days, cell numbers were counted. All experiments were performed three times, and mean values with standard deviations are indicated. *, P < 0.05.

Comments 4: Assess if Ser731Ala mutation affects tumor growth in vivo (xenograft models), which would bolster translational relevance.

Response 4: Thank you for your important suggestion. It would be very interesting to clarify whether the Ser731Ala mutation affects tumor growth in vivo. However, it is beyond the scope of this study. As a next step, we are considering testing tumor growth with the Ser731 mutation in a xenograft model.

Comments 5: Explore how PHF8-KDM2A dissociation enables rRNA suppression: Does freed KDM2A recruit repressive complexes to rDNA loci? ChIP-seq of KDM2A/PHF8 at rRNA genes under 2-DG treatment could elucidate this.

Response 5: Thank you for your suggestion. KDM2A binds to PHF8 through B region in addition to A region, and the wild type KDM2A fragment and the fragment retaining B region remain bound to PHF8 even after 2DG treatment (Figure 3A). We have previously shown that KDM2A binds to rRNA gene promoter via its CXXC-ZF domain (Tanaka et al., CxxC-ZF Domain Is Needed for KDM2A to Demethylate Histone in rDNA Promoter in Response to Starvation., Cell Struct. Funct. 2014), and the amount of KDM2A bound to the rRNA gene promoter does not change after starvation (Tanaka et al., JmjC enzyme KDM2A is a regulator of rRNA transcription in response to starvation., EMBO J., 2010; Tanaka et al., Mild glucose starvation induces KDM2A-mediated H3K36me2 demethylation through AMPK to reduce rRNA transcription and cell proliferation., Mol Cell Biol. 2015). While PHF8 is recruited to rRNA gene by binding to the active histone marks H3K4me3, the level of H3K4me3 does not change during starvation (Tanaka et al., JmjC enzyme KDM2A is a regulator of rRNA transcription in response to starvation., EMBO J., 2010). These results suggest that two proteins continuously present on rRNA gene promoter after 2DG treatment, but the binding mode between PHF8 and KDM2A may change under starvation. It is possible that A region from which PHF8 is released on starvation binds another molecule to reduce rRNA transcription. This binding protein may include components of repressive complexes. Alternatively, dissociation may affect histone demethylase activities of KDM2A and PHF8. Further studies are needed to clarify these points as future topics. To explain the possible mechanism for the reduction of rRNA transcription by PHF8-A region of KDM2A dissociation, the following sentences is added in line 574 in discussion section.

Text: The mechanism by which the dissociation between PHF8-A region of KDM2A allow rRNA suppression is an important question. KDM2A binds to PHF8 through the B region,in addition to the A region, and the wild type KDM2A fragment and the fragment retain-ing the B region remain bound to PHF8 after 2DG treatment (Figure 3A). We previously showed that KDM2A binds to rRNA gene promoter through CXXC-ZF domain [9], and the amount of KDM2A bound to the rDNA promoter does not change after starvation [8,23]. While PHF8 is recruited to rRNA gene by binding to the active histone marks H3K4me3, the level of H3K4me3 remained unchanged during starvation [8]. These results suggest that the two proteins continuously remain present on the rRNA gene promoter after 2DG treatment, but the binding mode between PHF8 and KDM2A may be altered by starvation. It is possible that the A region released from PHF8 on starvation binds another molecule and reduce rRNA transcription. This putative binding protein might contain a component of repressive complexes. Alternatively, the dissociation may affect the histone demethylase activities of KDM2A and PHF8. Further studies are needed to clarify these points.

Comments 6: Determine if other AMPK downstream effectors (e.g., TSC2) cross-talk with this pathway.

Response 6: Thank you for the comments on cross-talk of other AMPK downstream effectors with this pathway. In mammals, mTORC1 phosphorylates several transcription factors, including TIF-IA, form the Pol I transcription initiation complex (Mayer et al., mTOR-dependent activation of the transcription factor TIF-IA links rRNA synthesis to nutrient availability. Genes Dev, 2004; Sukumaran et al., Insight on Transcriptional Regulation of the Energy Sensing AMPK and Biosynthetic mTOR Pathway Genes. Front Cell Dev Biol, 2020; Huang et al., The TSC1-TSC2 complex: a molecular switchboard controlling cell growth. Biochem J , 2008). Thus, AMPK signaling reduces Pol I activity through TSC2-mediated TIF1A inhibition. While treatment of cells with 2 mM 2DG, under mild starvation condition, did not decrease the amount of TIF-IA in the rDNA promoter, this treatment decreased rRNA transcription in a KDM2A-dependent manner (Tanaka et al., Mild Glucose Starvation Induces KDM2A-Mediated H3K36me2 Demethylation through AMPK To Reduce rRNA Transcription and Cell Proliferation. Mol Cell Biol, 2015). These results suggest that KDM2A0mediated reduction in rRNA transcription does not require the AMPK- regulation of TIF-1A. At present, the molecular mechanisms by which KDM2A is dephosphorylated and how dephosphorylated KDM2A reduces rRNA transcription are not clear, and cross talk between AMPK downstream effectors has not been identified. Sentences have been added in discussion section in line 589 to explain the possible crosstalk between AMPK downstream effectors.

Text: In mammals, mTORC1 phosphorylates some transcription factors, including TIF-IA, for the formation of rRNA polymerase I (Pol I) transcription initiation complex [50-52]. AMPK phosphorylates TSC2 resulting in inhibition of mTORC1 [52], and TIF-IA binding to the rRNA gene promoter is reduced, and rRNA transcription is decreased [23,53]. Therefore, one way to reduce rRNA transcription by AMPK is that AMPK signaling re-duces Pol I activity through TSC2-mediated TIF1A inhibition. Although treatment of cells with mild starvation decreased rRNA transcription in a KDM2A-dependent manner, this treatment did not decrease the levels of TIF-IA in the rRNA gene promoter [23]. These re-sults suggest that KDM2A-mediated reduction of rRNA transcription does not require TIF-1A regulation by AMPK. So far, the molecular mechanisms by which KDM2A is dephosphorylated and how dephosphorylated KDM2A reduces rRNA transcription are not clear, and the cross talk between the AMPK downstream effectors has not been identi-fied.

Comments 7: Discuss if Ser731 is conserved across species or KDM2A paralogs (KDM2B).

Response 7: Thank you for your suggestion. It is very important to know whether important amino acids are conserved across species or KDM2A paralogs (KDM2B). Conservation of a protein’s amino acid sequences between species or within family proteins provides clues as to whether the sequence is important for functions. Comparison of the amino acid sequence around Ser731 (amino acid 700-808 of IDR-4) of human KDM2A with mouse and chicken sequences shows clear conservation between species. The level of identity in the sequences including Ser731 is similar to that of the JmjC domain (table 2). These results highlight the important function of this region and support the importance of the regulatory mechanism identified here. On the other hand, this conserved seine is absent in KDM2B (NP_115979.3), a KDM2A paralog. The lacking of the conserved serine would suggest that KDM2B does not respond to starvation as KDM2A. We added the explanation to line 556 of the discussion.

Text: The conservation of amino acid sequences in proteins between species or within family proteins provides clues as to whether the sequences are important to the functions. When the amino acid sequence around Ser731 of human KDM2A (amino acid 700-808 in IDR-4) was compared to those of mouse and chicken, the clear conservation was identified be-tween the species (99% identical between human and mouse and 91.7% identical between human and chicken). The identical levels in sequences containing Ser731 are similar to those of the sequences of the JmjC domain (table 2). These results highlight that this region has an important function and support the importance of the regulatory mechanism identified here. On the contrary, this conserved serine is absent in KDM2B (NP_115979.3), a KDM2A paralog. The lacking of the conserved serine would suggest that KDM2B does not respond to starvation as KDM2A.

Table 2.   Identity to human KDM2A (amino acids 1162) amino acids sequence (%)

The amino acid sequence for human (Homo sapience) KDM2A was taken from NP_036440.1, that for Mouse (Mus musculus) KDM2A was taken from NP_001001984.2, and that for Chicken (Gallus gallus) KDM2A was taken from XP_015128112.1.

Comments 8: Please use the standard three-line form for all the tales in the manuscript.

Response 8: We re-write tables, using the standard three-line form.

Comments 9: Gene or nucleotide sequence names should be italicized in the manuscript.

Response 9 Thank you for the suggestion. We italicized Gene or nucleotide sequence names in the manuscript.

Comments 10: Please add a scale bar to the microscope picture taken in the text.

Response 10: Thank you for the suggestion. We added scale bars in the pictures (Figure 3A).

Comments 11: Abbreviations should be defined when they first appear in the text. Such as "PHF8" in Line 24, "SDS-PAGE" in Line 122, ...please double check all the text and revsie them.???

Response 11: Thank you for the suggestion. We double checked if abbreviations were defined when they first appeared in the text.

Comments 12: Please unify the format of references in the article, including the author's name, the case of words in the title of the article, the writing of the name of the journal, and the page number.

Response 12: Thank you for the suggestion. We double checked references.

Reviewer 1

Attached please find our revised manuscript entitled “Interaction between PHF8 and a segment of KDM2A, which is controlled by the phosphorylation status at a specific serine in an intrinsically disordered region of KDM2A, regulates rRNA transcription and cell proliferation in a breast cancer cell line”, which we are re-submitting to your consideration for publication in Biomolecules as an article. The manuscript was revised following the useful comments we received from the reviewers. We could respond to all the comments. We added the results of the new experiments in the revised version of the manuscript. The newly added figures are Figure S1, Figure S3, Figure S9, and the newly added table is table 2. The improved figures are Figure 3A, in which we added scale bars. Our conclusion could be far clearly stated in this new version.

 Point-by-point response to Comments (Referee #1)

Comments 1: Direct evidence showing AMPK phosphorylates/dephosphorylates KDM2A at Ser731 (e.g., in vitro kinase assays with AMPK and KDM2A peptides) would strengthen the central claim

Response 1: Thank you for your suggestion. The relationship between AMPK and regulation of the phosphorylation state of Ser731 in KDM2A is an important issue to address. Phosphorylated Ser731 was detected when KDM2A was expressed in 293T cells. Phosphorylated Ser731 was decreased by the addition of 2DG or the AMPK activator AICAR, and the decrease was prevented by the addition of cantharidin, a phosphatase inhibitor that strongly and selectively inhibits protein phosphatase 2A (PP2A). These results suggest that PP2A is activated by AMPK, which dephosphorylates KDM2A at Ser731. To illustrate these results, a new figure has been added as Figure S9 and is discussed in line 609, in discussion section as below.

Text:

When KDM2A was expressed in 293T cells, phosphorylated Ser731 was detected (Figure S9). Addition of 2DG or AMPK activator AICAR reduced the level of phosphorylated Ser731, and the potent and selective inhibitor of protein phosphatase 2A (PP2A), the phosphatase inhibitor cantharidin, prevented the reduction of pSer731 levels (Figure S9). These results suggest that PP2A is activated downstream of the AMPK pathway, and dephosphorylates KDM2A at Ser731.  

Figure S9. Evidence for phosphatase involvement in dephosphorylation of Ser731 of KDM2A on starvation. An expression vector encoding KDM2A was transfected into 293T cells by electroporation, and then siRNA for KDM2A was transfected. After cultured for 2 d, cells were treated with a phosphatase inhibitor, cantharidin, at the indicated concentrations, for 4 h in the presence or absence of 2mM 2DG (A) or 0.5 mM AICAR (B). Cells were collected and analyzed by Western blotting using anti-phosphorylated Ser731 antibody and anti-KDM2A antibody (Abcam, ab191387) (upper panel). The relative phosphorylation levels were expressed as the values of phosphorylated Ser731 divided by the values for KDM2A. The value without 2DG (A) or AICAR (B), and cantharidin was expressed as 1 (lower panel).

Comments 2: Include rescue experiments (e.g., AMPK knockdown + phospho-mimetic Ser731 mutants) to confirm the AMPK-dependence of Ser731 modification.

Response 2: As indicated in response to Question #1, experiments with the PP2A inhibitor cantharidin suggested that AMPK activates PP2A to reduce pS731 levels, supporting an AMPK-dependent nature of Ser731 modification. To illustrate these results, a new figure has been added as Figure S9 and is explained in line 609 of the Discussion section, as answered in response to Reviewer #1's Question 1.

Comments 3: Test whether this mechanism is conserved in other breast cancer subtypes (e.g., TNBC) or glucose-starvation models (e.g., low-glucose media).

Response 3: Thank you for pointing this out. We tested whether PHF8 is involved in the reduction of rRNA transcription by 2DG treatment in the TNBC cell line, MDA-MB-231. The results showed the involvement of PHF8 in response to starvation in TNBC cell line as well. The results were described in line 257 of text, and a new figure is added as Figure S3.  

Text: We also tested whether PHF8 was involved in reduction of rRNA transcription in TNBC cell line MDA-MB-231 on 2DG treatment. The reductions of rRNA transcription and cell numbers were suppressed by either KDM2A or PHF8 KD in MDA-MB-231 cells (Figure S3). These results suggest that PHF8 and KDM2A are involved in the reduction of rRNA transcription and cell proliferation in breast cancer cells.

Figure S3. Effects of PHF8 and KDM2A knockdown on TNBC cell line MDA-MB-231. (A) MDA-MB-231 cells (European collection of authenticated cell cultures, ECACC, Catalogue No. 92020424) were transfected with stealth siRNAs for KDM2A or PHF8 (siPHF8#1 and siPHF8#3), and cultured for 3 d. The oligonucleotide sequence of stealth siRNA siPHF8#3 is 5’-CAUUCCACUUCAGUGUCCAUGUCCA-3’. Cells were replated in the growth medium. The next day, cells were cultured in the presence or absence of 3 mM 2DG for 2 h. Total RNA was isolated and pre-rRNA, KDM2A mRNA, and PHF8 mRNA were detected by RT-PCR. Results were normalized by Polr2a mRNA value. (B) Involvement of KDM2A and PHF8 in inhibition of cell proliferation by 2DG. Cells transfected as in (A) were replated and cultured in medium in the presence or absence of 3 mM 2DG. On the indicated days, cell numbers were counted. All experiments were performed three times, and mean values with standard deviations are indicated. *, P < 0.05.

Comments 4: Assess if Ser731Ala mutation affects tumor growth in vivo (xenograft models), which would bolster translational relevance.

Response 4: Thank you for your important suggestion. It would be very interesting to clarify whether the Ser731Ala mutation affects tumor growth in vivo. However, it is beyond the scope of this study. As a next step, we are considering testing tumor growth with the Ser731 mutation in a xenograft model.

Comments 5: Explore how PHF8-KDM2A dissociation enables rRNA suppression: Does freed KDM2A recruit repressive complexes to rDNA loci? ChIP-seq of KDM2A/PHF8 at rRNA genes under 2-DG treatment could elucidate this.

Response 5: Thank you for your suggestion. KDM2A binds to PHF8 through B region in addition to A region, and the wild type KDM2A fragment and the fragment retaining B region remain bound to PHF8 even after 2DG treatment (Figure 3A). We have previously shown that KDM2A binds to rRNA gene promoter via its CXXC-ZF domain (Tanaka et al., CxxC-ZF Domain Is Needed for KDM2A to Demethylate Histone in rDNA Promoter in Response to Starvation., Cell Struct. Funct. 2014), and the amount of KDM2A bound to the rRNA gene promoter does not change after starvation (Tanaka et al., JmjC enzyme KDM2A is a regulator of rRNA transcription in response to starvation., EMBO J., 2010; Tanaka et al., Mild glucose starvation induces KDM2A-mediated H3K36me2 demethylation through AMPK to reduce rRNA transcription and cell proliferation., Mol Cell Biol. 2015). While PHF8 is recruited to rRNA gene by binding to the active histone marks H3K4me3, the level of H3K4me3 does not change during starvation (Tanaka et al., JmjC enzyme KDM2A is a regulator of rRNA transcription in response to starvation., EMBO J., 2010). These results suggest that two proteins continuously present on rRNA gene promoter after 2DG treatment, but the binding mode between PHF8 and KDM2A may change under starvation. It is possible that A region from which PHF8 is released on starvation binds another molecule to reduce rRNA transcription. This binding protein may include components of repressive complexes. Alternatively, dissociation may affect histone demethylase activities of KDM2A and PHF8. Further studies are needed to clarify these points as future topics. To explain the possible mechanism for the reduction of rRNA transcription by PHF8-A region of KDM2A dissociation, the following sentences is added in line 574 in discussion section.

Text: The mechanism by which the dissociation between PHF8-A region of KDM2A allow rRNA suppression is an important question. KDM2A binds to PHF8 through the B region,in addition to the A region, and the wild type KDM2A fragment and the fragment retain-ing the B region remain bound to PHF8 after 2DG treatment (Figure 3A). We previously showed that KDM2A binds to rRNA gene promoter through CXXC-ZF domain [9], and the amount of KDM2A bound to the rDNA promoter does not change after starvation [8,23]. While PHF8 is recruited to rRNA gene by binding to the active histone marks H3K4me3, the level of H3K4me3 remained unchanged during starvation [8]. These results suggest that the two proteins continuously remain present on the rRNA gene promoter after 2DG treatment, but the binding mode between PHF8 and KDM2A may be altered by starvation. It is possible that the A region released from PHF8 on starvation binds another molecule and reduce rRNA transcription. This putative binding protein might contain a component of repressive complexes. Alternatively, the dissociation may affect the histone demethylase activities of KDM2A and PHF8. Further studies are needed to clarify these points.

Comments 6: Determine if other AMPK downstream effectors (e.g., TSC2) cross-talk with this pathway.

Response 6: Thank you for the comments on cross-talk of other AMPK downstream effectors with this pathway. In mammals, mTORC1 phosphorylates several transcription factors, including TIF-IA, form the Pol I transcription initiation complex (Mayer et al., mTOR-dependent activation of the transcription factor TIF-IA links rRNA synthesis to nutrient availability. Genes Dev, 2004; Sukumaran et al., Insight on Transcriptional Regulation of the Energy Sensing AMPK and Biosynthetic mTOR Pathway Genes. Front Cell Dev Biol, 2020; Huang et al., The TSC1-TSC2 complex: a molecular switchboard controlling cell growth. Biochem J , 2008). Thus, AMPK signaling reduces Pol I activity through TSC2-mediated TIF1A inhibition. While treatment of cells with 2 mM 2DG, under mild starvation condition, did not decrease the amount of TIF-IA in the rDNA promoter, this treatment decreased rRNA transcription in a KDM2A-dependent manner (Tanaka et al., Mild Glucose Starvation Induces KDM2A-Mediated H3K36me2 Demethylation through AMPK To Reduce rRNA Transcription and Cell Proliferation. Mol Cell Biol, 2015). These results suggest that KDM2A0mediated reduction in rRNA transcription does not require the AMPK- regulation of TIF-1A. At present, the molecular mechanisms by which KDM2A is dephosphorylated and how dephosphorylated KDM2A reduces rRNA transcription are not clear, and cross talk between AMPK downstream effectors has not been identified. Sentences have been added in discussion section in line 589 to explain the possible crosstalk between AMPK downstream effectors.

Text: In mammals, mTORC1 phosphorylates some transcription factors, including TIF-IA, for the formation of rRNA polymerase I (Pol I) transcription initiation complex [50-52]. AMPK phosphorylates TSC2 resulting in inhibition of mTORC1 [52], and TIF-IA binding to the rRNA gene promoter is reduced, and rRNA transcription is decreased [23,53]. Therefore, one way to reduce rRNA transcription by AMPK is that AMPK signaling re-duces Pol I activity through TSC2-mediated TIF1A inhibition. Although treatment of cells with mild starvation decreased rRNA transcription in a KDM2A-dependent manner, this treatment did not decrease the levels of TIF-IA in the rRNA gene promoter [23]. These re-sults suggest that KDM2A-mediated reduction of rRNA transcription does not require TIF-1A regulation by AMPK. So far, the molecular mechanisms by which KDM2A is dephosphorylated and how dephosphorylated KDM2A reduces rRNA transcription are not clear, and the cross talk between the AMPK downstream effectors has not been identi-fied.

Comments 7: Discuss if Ser731 is conserved across species or KDM2A paralogs (KDM2B).

Response 7: Thank you for your suggestion. It is very important to know whether important amino acids are conserved across species or KDM2A paralogs (KDM2B). Conservation of a protein’s amino acid sequences between species or within family proteins provides clues as to whether the sequence is important for functions. Comparison of the amino acid sequence around Ser731 (amino acid 700-808 of IDR-4) of human KDM2A with mouse and chicken sequences shows clear conservation between species. The level of identity in the sequences including Ser731 is similar to that of the JmjC domain (table 2). These results highlight the important function of this region and support the importance of the regulatory mechanism identified here. On the other hand, this conserved seine is absent in KDM2B (NP_115979.3), a KDM2A paralog. The lacking of the conserved serine would suggest that KDM2B does not respond to starvation as KDM2A. We added the explanation to line 556 of the discussion.

Text: The conservation of amino acid sequences in proteins between species or within family proteins provides clues as to whether the sequences are important to the functions. When the amino acid sequence around Ser731 of human KDM2A (amino acid 700-808 in IDR-4) was compared to those of mouse and chicken, the clear conservation was identified be-tween the species (99% identical between human and mouse and 91.7% identical between human and chicken). The identical levels in sequences containing Ser731 are similar to those of the sequences of the JmjC domain (table 2). These results highlight that this region has an important function and support the importance of the regulatory mechanism identified here. On the contrary, this conserved serine is absent in KDM2B (NP_115979.3), a KDM2A paralog. The lacking of the conserved serine would suggest that KDM2B does not respond to starvation as KDM2A.

Table 2.   Identity to human KDM2A (amino acids 1162) amino acids sequence (%)

The amino acid sequence for human (Homo sapience) KDM2A was taken from NP_036440.1, that for Mouse (Mus musculus) KDM2A was taken from NP_001001984.2, and that for Chicken (Gallus gallus) KDM2A was taken from XP_015128112.1.

Comments 8: Please use the standard three-line form for all the tales in the manuscript.

Response 8: We re-write tables, using the standard three-line form.

Comments 9: Gene or nucleotide sequence names should be italicized in the manuscript.

Response 9 Thank you for the suggestion. We italicized Gene or nucleotide sequence names in the manuscript.

Comments 10: Please add a scale bar to the microscope picture taken in the text.

Response 10: Thank you for the suggestion. We added scale bars in the pictures (Figure 3A).

Comments 11: Abbreviations should be defined when they first appear in the text. Such as "PHF8" in Line 24, "SDS-PAGE" in Line 122, ...please double check all the text and revsie them.???

Response 11: Thank you for the suggestion. We double checked if abbreviations were defined when they first appeared in the text.

Comments 12: Please unify the format of references in the article, including the author's name, the case of words in the title of the article, the writing of the name of the journal, and the page number.

Response 12: Thank you for the suggestion. We double checked references.

Reviewer 2 Report

Comments and Suggestions for Authors

The paper is interesting, the authors have done quite extensive cell culture work, however it needs modifications, and the below questions need to be answered. 

Please give a good introduction to PHF8. What is the protein's full name?

pre-Ribosomal RNA transcription and processing is a highly orchestrated process. The authors mentioned in the abstract that KDM2A binds to the promoter region of the rRNA gene. Using a reductionist approach, a binding assay with synthetic DNA that represents the promoter sequence of rRNA gene and with purified KDM2A and PHF8 will add strength to the paper. 

A few key points needs to be addressed. 

Does KDM2A and PHF8 have a nuclear import signal? If so, please specify the domain in the introduction?

Can the IP be replicated in another breast cancer cell line other than MCF-7? Can IP be done with just nuclear extract and/or western blots normalized to a nuclear protein like lamin b1, instead of highly abundant b-actin?

Please provide details and justification of selection of primer for pre-rRNA. Is it the small subunit or large subunit? From what I know pre-rRNA do not have a polyA tail, so reverse transcription for qRT-PCR requires special primers to amplify pre-rRNA and would be more accurate with RNA prepared from nuclear extract. 

What was the criteria for selection of cell lysis buffer for immunoprecipitation [0.1% Triton X-100, 300 mM NaCl, 300 mM sucrose, 1 mM 126 MgCl2, 1 mM EGTA, and 10 mM PIPES (pH 7.0) ] Since these are nuclear proteins, what levels of KDM2A and PHF8 were obtained in the pellet after lysis and how do they compare with the levels in the supernatant?

Author Response

Reviewer 2

Attached please find our revised manuscript entitled “Interaction between PHF8 and a segment of KDM2A, which is controlled by the phosphorylation status at a specific serine in an intrinsically disordered region of KDM2A, regulates rRNA transcription and cell proliferation in a breast cancer cell line”, which we are re-submitting to your consideration for publication in Biomolecules as an article. The manuscript was revised following the useful comments we received from the reviewers. We could respond to all the comments. We added the results of the new experiments in the revised version of the manuscript. The newly added figures are Figure S1, Figure S3, Figure S9, and the newly added table is table 2. The improved figures are Figure 3A, in which we added scale bars. Our conclusion could be far clearly stated in this new version.

 Point-by-point replies to Referee #2:

Comments 1: Please give a good introduction to PHF8. What is the protein's full name?

Response 1 : Thank you for your suggestion. We added the full name of PHF8 (PHD finger protein 8), to line 24 of the abstract. To introduce PHF8, we added information about PHF8 to line 88 of the introduction.  This information describes the putative NLSs, binding to histone marks, and regulation of rRNA transcription as follows.

Text: PHF8 acts as a demethylase for the repressive histone marks (33). PHF8 has six putative nuclear localization signals in its amino acids sequence (34). PHF8 binds the active his-tone marks through its PHD. Recombinant PHF8 bound to a synthetic histone H3 peptides comprising H3K4me3 and, to a lesser extent, to H3K4me2, but did not interact with un-modified histone H3 peptide, H3K4me1, H3K9me2 or H3K9me3 (35). These results suggest that PHF8 is anchored to the active rDNA repeats that are demarcated by H3K4me3. PHF8 up-regulates rRNA transcription by reducing the repressive histone marks (H3K9me2) through its demethylase activity (35, 36). However, whether PHF8 activity is altered dur-ing starvation has not yet been tested.

Comments 2: pre-Ribosomal RNA transcription and processing is a highly orchestrated process. The authors mentioned in the abstract that KDM2A binds to the promoter region of the rRNA gene. Using a reductionist approach, a binding assay with synthetic DNA that represents the promoter sequence of rRNA gene and with purified KDM2A and PHF8 will add strength to the paper. 

Response 2: Thank you for your suggestion. We have already reported that KDM2A binds to rRNA gene promoter sequence in vitro (Tanaka et al., CxxC-ZF Domain Is Needed for KDM2A to Demethylate Histone in rDNA Promoter in Response to Starvation., Cell Structure and Functio., 2014). We added the important information about binding of KDM2A to rRNA gene in line 48 of the introduction as follows. Another group has already reported that PHF8 binds to the active histone marks. As mentioned in the answer to question 1, we added the information PHF8 binding to the active histone marks in line 88.

Text: Because the rRNA gene promoter contains lots of CpG dinucleotides (11, 12) and CxxC-ZF domain binds to unmethylated CpG (11, 13-15), we previously tested whether the CxxC-ZF domain was involved in binding of KDM2A to the rRNA gene promoter (9). The recombinant GST-fusion protein containing CxxC-ZF domain of KDM2A binds to the rRNA gene promoter depending on unmethylated state of DNA and the integrity of CxxC-ZF domain in vitro. The binding of KDM2A to rRNA gene promoter in cells was also observed by ChIP analysis in vivo. These results indicate that KDM2A is recruited to the rRNA gene pro-moter depending on the CxxC-ZF domain.

Comments 3. A few key points needs to be addressed.  Does KDM2A and PHF8 have a nuclear import signal? If so, please specify the domain in the introduction?

Response 3: Thank you for your suggestion. In the case of KDM2A, at least to our knowledge no nuclear import signal was reported. Instead, we searched nucleolar localization signals and reported three regions were involved in nucleolar localization of KDM2A: KDM2A (amino acids 543–666), KDM2A (amino acids 768–819), KDM2A (amino acids 947–1162). Thus, we added information about the nucleolar localization signals of KDM2A to line 80 of the introduction. In the case of PHF8, six putative NLS sequences are predicted as described in the answer to question 1 (added in text to line 88 of introduction in the main text).

Text: To investigate the mechanisms that regulate the activity of KDM2A in the nucleolus, we searched previously amino acid sequence involved in nucleolar localization [24]. The results showed that there are multiple regions involved in nucleolar localization of KDM2A. One region contains CxxC-ZF, which directly binds to rRNA gene promoter [9], and another region binds directly to heterochromatin protein 1γ (HP1γ) using valine 801 in the LxVxL motif of KDM2A [24]. Both regions were required for KDM2A to suppress rRNA transcription during starvation. [9,24].

Comments 4: Can the IP be replicated in another breast cancer cell line other than MCF-7? Can IP be done with just nuclear extract and/or western blots normalized to a nuclear protein like lamin b1, instead of highly abundant b-actin?

Response 4: Thank you for your suggestion. Previously, we detected KDM2A by Western blotting in TNBC cell line MDA-MB-231 (Okamoto et al., KDM2A-dependent reduction of rRNA transcription on glucose starvation requires HP1 in cells, including triple-negative breast cancer cells. Oncotarget 2019; Tanaka et al., Mild Glucose Starvation Induces KDM2A-Mediated H3K36me2 Demethylation through AMPK To Reduce rRNA Transcription and Cell Proliferation, MCB 2015). In this study, we detected endogenous and ectopically expressed KDM2A and PHF8 in 293T cell extracts (see Figure S1, answer to question 6 by Reviewers #2, although these cells were not breast cancer cells).

As normalization factors, we detected two nuclear proteins, nucleolin and nucleophosmin (B23), in addition to β-actin. When these proteins were used as normalization factors, the reduction rates of pS731/KDM2A by 2DG were very similar to that when β-actin was used as a normalization factor, suggesting that β-actin can be used to evaluate pS731/KDM2A as a normalization factor. Nucleolin and nucleophosmin (B23) were not used as normalization factors in most experiments, so we have attached the results for the reviewers in this letter. (Reviewer Figure)

Reviewer Figure. MCF-7 cells were replated and cultured for 2 h in the presence or absence of 2DG. Proteins were extracted and the levels of pS731 of KDM2A, KDM2A, β-actin, nucleolin, and nucleophosmin (B23) were detected by western blotting using the specific antibodies. Nucleolin was detected by anti-nucleolin antibody (Santa Cruz Biotechnology, C23 (H250), sc13057). Nucleophospmin (B23) was detected by anti-nucleophismine specific antibody (Santa Cruz Biotechnology, B23 (H106), sc5564) (left-side panel). Band intensitiy was measured, and pS731 levels were divided by KDM2A level. The reduction rates was normalized toβ-actin, nucleolin, or nucleophosmin (B23) level. Normalized relative reduction rates are presented (right-hand panel).

Comments 5: Please provide details and justification of selection of primer for pre-rRNA. Is it the small subunit or large subunit? From what I know pre-rRNA do not have a polyA tail, so reverse transcription for qRT-PCR requires special primers to amplify pre-rRNA and would be more accurate with RNA prepared from nuclear extract. 

Response 5: Thank you for your suggestion. To measure rRNA transcription, we amplified the sequence immediately following the transcription start site (1-155) in the external transcribed spacer (ETS). This sequence does not include 18S or 28S rRNA. The RNA in this region is destroyed immediately after transcription, so the amounts of RNA reflect rRNA transcription level. To detect the rRNA transcription level, we isolated total RNA from cells, and synthesized single-strand cDNA from the total RNA using random primers (Tanaka et al., JmjC enzyme KDM2A is a regulator of rRNA transcription in response to starvation., EMBO J., 2010). Using this protocol we successfully amplified ETS of pre-rRNA, 28S rRNA, and transcripts produced by Pol II, including KDM2A and PHF8. To clarify the method for detection of the transcripts, we added the following explanation to line 207, in materials and methods section.

Text: Isolation of total RNA from cells and cDNA synthesis were performed as described previously (8, 24). In brief, total RNA was isolated from cells, and single-strand cDNA was synthesized using random primers. The products were diluted to 150 μl with distilled water, and 2.5 μl of the resultant single-strand cDNA was used as the template for qRT-PCR, using a KAPA SYBR Fast qPCR kit (NIPPON Genetics, Tokyo, Japan) with a CSF ConnectTM Real-Time PCR Detection System (Bio-Rad Laboratories, Inc., Hercules, CA) according to the manufacturer’s instructions (8). The sets of PCR primers for amplification were shown below (PCR primers). To measure rRNA transcription, we amplified sequence immediately after the transcription start site (1-155) in external transcribed spacer (ETS). The RNA in this region is destroyed soon after transcribed and the amounts of the RNA reflect rRNA transcription levels (8).

Comments 6: What was the criteria for selection of cell lysis buffer for immunoprecipitation [0.1% Triton X-100, 300 mM NaCl, 300 mM sucrose, 1 mM 126 MgCl2, 1 mM EGTA, and 10 mM PIPES (pH 7.0) ] Since these are nuclear proteins, what levels of KDM2A and PHF8 were obtained in the pellet after lysis and how do they compare with the levels in the supernatant?

Response 6: Thank you for your suggestion. We investigated the extraction levels with IP buffer (0.1% Triton X-100, 300 mM NaCl, 300 mM sucrose, 1 mM MgCl2, 1 mM EGTA, and 10 mM PIPES (pH 7.0)). Both KDM2A and PHF8 were almost completely extracted with IP buffer. The protein levels in the supernatant of IP buffer extraction were similar to those extracted by 3%SDS solution. No protein was detected in the precipitated fraction of IP buffer extraction. These results indicate that the two proteins were well extracted by IP buffer. The extraction mode of β-actin was similar to that of KDM2A and PHF8. On the other hand, histone H3 was mainly collected in the precipitated fraction. We added Figure S1 and an explanation on line 230 of the results, to show that the extraction of PHF8 and KDM2A was successful.  

Text: First, the levels of extraction of proteins by IP buffer (0.1% Triton X-100, 300 mM NaCl, 300 mM sucrose, 1 mM MgCl2, 1 mM EGTA, and 10 mM PIPES, pH 7.0) was investigated. Both KDM2A and PHF8 were almost completely extracted by IP buffer (Figure S1). The protein levels in supernatant of IP buffer extraction were similar to those extracted by 3% SDS soln. No proteins (KDM2A and PHF8) were detected in the precipitated fractions of IP buffer ex-traction. The extraction mode of β-actin was similar to those of KDM2A and PHF8. On the other hand, histone H3 was mainly recovered in the precipitated fraction. These results indicate that KDM2A and PHF8 are well extracted in IP buffer.

Figure S1. Extraction of KDM2A and PHF8 from cells using IP buffer. (A) 293T cells were transfected with an empty vector or a vector expressing Flag-PHF8 and a vector expressing KDM2A by electroporation. After two days of culture, the cells were harvested using trypsin and EDTA, and extracted with 3% SDS soln (Whole), or immunoprecipitation buffer (IP buffer). Proteins extracted with IP buffer were centrifuged at 20,000 g and separated into supernatant fraction (Sup) and precipitated fraction (Ppt). The 3% SDS soln were added to the collected samples, and samples prepared from the same number of cells were analyzed by western blotting using anti-PHF8 antibody, anti-KDM2A antibody (rabbit anti-KDM2A monoclonal antibody (Abcam, ab191387)), anti-β-actin antibody, and anti-histone H3 antibody (rabbit polyclonal anti-histone H3 antibody, Abcam, ab1791) (upper panels). Band intensities were measured and relative amounts were shown (lower panels).

Round 2

Reviewer 1 Report

Comments and Suggestions for Authors

The authors have addressed all my comments, I recommend accepting it in current form.

Reviewer 2 Report

Comments and Suggestions for Authors

Thank you for your response.